# A novel Fer/FerT targeting compound selectively evokes metabolic stress and necrotic death in malignant cells

Yoav Elkis[1], Moshe Cohen[1], Etai Yaffe[1], Shirly Satmary-Tusk[1], Tal Feldman[1], Elad Hikri[1], Abraham Nyska[2], Ariel Feiglin[1], Yanay Ofran[1], Sally Shpungin[1] & Uri Nir[1]

Disruption of the reprogrammed energy management system of malignant cells is a prioritized goal of targeted cancer therapy. Two regulators of this system are the Fer kinase, and its cancer cell specific variant, FerT, both residing in subcellular compartments including the mitochondrial electron transport chain. Here, we show that a newly developed inhibitor of Fer and FerT, E260, selectively evokes metabolic stress in cancer cells by imposing mitochondrial dysfunction and deformation, and onset of energy-consuming autophagy which decreases the cellular ATP level. Notably, Fer was also found to associate with PARP-1 and E260 disrupted this association thereby leading to PARP-1 activation. The cooperative intervention with these metabolic pathways leads to energy crisis and necrotic death in malignant, but not in normal human cells, and to the suppression of tumors growth in vivo. Thus, E260 is a new anti-cancer agent which imposes metabolic stress and cellular death in cancer cells.

[1] The Mina and Everard Goodman Faculty of Life-Sciences, Bar-Ilan University, Ramat-Gan 52900, Israel. [2] Consultant in Toxicological Pathology, Timrat, and Sackler School of Medicine, Tel Aviv University, Tel Aviv 36576, Israel. Yoav Elkis, Moshe Cohen and Etai Yaffe contributed equally to this work. Correspondence and requests for materials should be addressed to U.N. (email: uri.nir@biu.ac.il)

Targeted therapy of cancer is aimed towards the development of selective inhibitors of the aberrant and mutated regulatory pathways of tumor cells, thereby leading to the elimination of malignant tumors. However, vast amounts of accumulating evidence highlight the complexity and challenging nature of this goal. This complexity reflects the genomic instability of malignant cells, and their tendency to acquire resistance to therapeutic agents[1]. To overcome these obstacles, a novel approach has been adopted based on targeting fundamental processes that characterize the reprogrammed metabolic and energy generation systems of cancer cells[2]. Specifically, while normal mammalian cells primarily utilize mitochondrial

oxidative phosphorylation for adenosine-tri-phosphate (ATP) production, cancer cells remodel their glycolytic and mitochondrial machinery so that glycolysis is upregulated even under aerobic conditions, which would normally attenuate glycolysis, a phenomenon termed the "Warburg effect"[3]. The enhanced glycolytic capability of malignant cells might be related to the overexpression of glycolytic enzymes such as hexokinase II (HK II), which is present only at basal levels in normal somatic cells and can facilitate the malignant phenotype[4]. HK II bears a double catalytic domain and is attached to the outer mitochondrial surface via the voltage-dependent anion channel, thereby enabling it to directly and efficiently utilize mitochondria-produced ATP to phosphorylate glucose at a faster rate[4]. Although the "Warburg effect" is a hallmark of the reprogrammed metabolism of cancer cells, these cells remain dependent on the integrity and functionality of their mitochondria for ATP production and fatty acid synthesis, a requirement that becomes most profound upon transition of the malignant disease to a metastatic phase[5]. Thus, the mitochondrial machinery undergoes reprograming during the development and progression of malignant disease, a change that is reflected in the altered activity of several key enzymes[6, 7].

A recently reported player in mitochondrial reprogramming in cancer cells is the intracellular tyrosine-kinase, Fer, and its sperm and cancer cell-specific truncated variant, FerT, which are harnessed to the reprogrammed mitochondria in colon carcinoma[8] cells[7]. Fer populates several subcellular compartments in malignant cells, including the cytoplasmic membrane, mitochondria, and cell nucleus[7, 9, 10]. In the mitochondria, Fer and FerT associate with complex I of the mitochondrial electron transport chain (ETC) of malignant but not of normal somatic cells, thereby supporting ATP production in nutrient-deprived cancer cells, in a kinase dependent manner[7]. Furthermore, silencing of either Fer or FerT is sufficient to impair ETC complex I activity. Concomitantly, directed mitochondrial accumulation of FerT in nonmalignant NIH3T3 cells increases their ETC complex I activity, ATP production, and survival, contingent upon stress conditions imposed by nutrient and oxygen deprivation. Notably, enforced mitochondrial expression of FerT endowed the non-malignant cells with an ability to form tumors in vivo[7]. Thus, recruitment of the meiotic FerT to cancer cell mitochondria highlights the primary role of reprogrammed mitochondria in tumorigenesis.

Several lines of evidence support the roles of Fer in the progression and growth of malignant tumors. The kinase was detected in all human malignant cell lines analyzed[11, 12] and its levels in malignant prostate tumors are significantly higher than those detected in benign growths/tumors[13]. Furthermore, downregulation of Fer impairs the proliferation of prostate, breast, and colon carcinoma[8] cells[10], induces death in CC and non-small cell lung cancer (NSCLC) cells[14, 15], abolishes the ability of prostate carcinoma PC3 and V-sis-transformed cells to form colonies in soft agar[13], and delays the onset and reduces the proliferation rate of mammary gland tumors in HER2 over-expressing transgenic mice[16]. Fer was also shown to promote metastatic processes; downregulation of Fer prevents the metastatic spread of breast and lung adenocarcinoma tumors[17, 18]. At the clinical level, high Fer expression levels have been linked to poor prognosis of hepatocellular-carcinoma (HCC)[19], clear cell renal cell carcinoma[20, 21], postoperative NSCLC[14], and high-grade basal/triple-negative breast cancer[22]. The above findings portray the Fer and FerT kinases, which share an identical kinase domain (KD)[23], as potential targets for selectively affecting the reprogrammed metabolic systems of cancer cells.

Although targeting cancer cell metabolism and mitochondria with synthetic compounds may prove useful, cancer cells can overcome metabolic insults by transiently inducing the autophagy salvage process[24]. Autophagy governs controlled cellular self-degradation, which is important for balancing sources of energy at critical points in response to metabolic insults[24]. This process also plays a cell-preserving role by removing and recycling damaged organelles, such as mitochondria, endoplasmic reticulum, and peroxisomes[25]. Though autophagy is considered to promote survival, it can also lead to a cellular death, like necrosis[26]. One of the drivers of autophagy and onset of necrotic death is the poly (ADP-ribose) polymerase-1 (PARP-1) enzyme. PARP-1 takes part in multiple cellular pathways, including the adenosine-mono-phosphate protein kinase (AMPK)–mammalian target of rapamycin (mTOR) signaling cascade[27], which controls autophagy, and deregulated activity of PARP-1 can lead to ATP and NAD consumption and to the onset of necrotic death[28].

In this manuscript, we describe a novel synthetic Fer/FerT inhibitor, which compromises mitochondrial integrity, and activates PARP-1 and autophagy, thereby targeting the energy management system of cancer cells, and selectively leading to their necrotic death.

## Results

**Development of a novel inhibitor of the Fer/FerT kinases**. To screen and develop a new and selective inhibitor of the Fer and FerT kinases, a yeast-based, high throughput screening (HTS) system was adopted. This HTS system is based on the previously described growth inhibition of yeast cells by ectopic expression of a mammalian kinase in these cells[29]. To optimize this system for the identification and development of Fer inhibitors, we ectopically expressed Fer in the yeast strain BY4741-3702, which lacks the phosphatase encoding-*ptc1*gene. Ectopic expression of Fer was found to change the tyrosine-phosphorylation pattern of intracellular yeast proteins (Fig. 1a) in a manner dependent on the Fer-kinase activity (Fig. 1a). Concomitantly, ectopic presence of active Fer attenuated the growth of the expressing yeast cells, an effect that was not seen in cells expressing an inactive Fer-mutant[23] (Fig. 1b). Thus, attenuated growth of the BY4741-3702

**Fig. 1** Development of the Fer/FerT kinase inhibitor-E260 by using a yeast-based HTS system. **a** Native Fer (pAES-Fer) and a mutated kinase inactive Fer (pAES-FerY715F) were expressed in yeast cells using the pAES expression vector. Protein lysates from transfected cells were subjected to WB analysis using anti-HA (for HA-tagged Fer), anti-Fer, anti-phosphotyrosine (p-tyr), and anti-actin antibodies. **b** Growth curve of yeast cells expressing either the empty pAES vector, native Fer, inactive Fer, and non-transformed yeast cells; $n = 3, \pm SE$. **c** Scheme of the HTS program. **d** The chemical structure the 0342 compound. **e** The chemical structure of the E260 compound. **f** Growth curve of yeast transformed with the different plasmids and treated with either DMSO, 2 μM E260, or 2 μM 0342. pAES-harboring cells treated with DMSO, mutant Fer treated with DMSO, native Fer-expressing cells treated with DMSO, native Fer-expressing cells treated with 0342, native Fer-expressing cells treated with E260. Untransformed yeast cells served as a negative control; $n = 3, \pm SE$. **g** Histograms depicting the 600 nm OD readings of the yeast growth-curves (presented in **f**) after 40 h; $n = 3, \pm SE$. **h** Protein lysates prepared from yeast expressing mutant Fer and treated with DMSO for 24 h (pAES-FerY715F), active Fer and treated with 2 μM E260 (pAES-Fer E260), active Fer and treated with DMSO (pAES-Fer DMSO), or harboring the pAES vector alone and treated with DMSO, were subjected to WB analysis using anti-Fer, anti-p-tyr, and anti-PGK antibodies. Arrow on the right depicts the MW of Fer. This image presents one experiment representative of three, which gave similar results

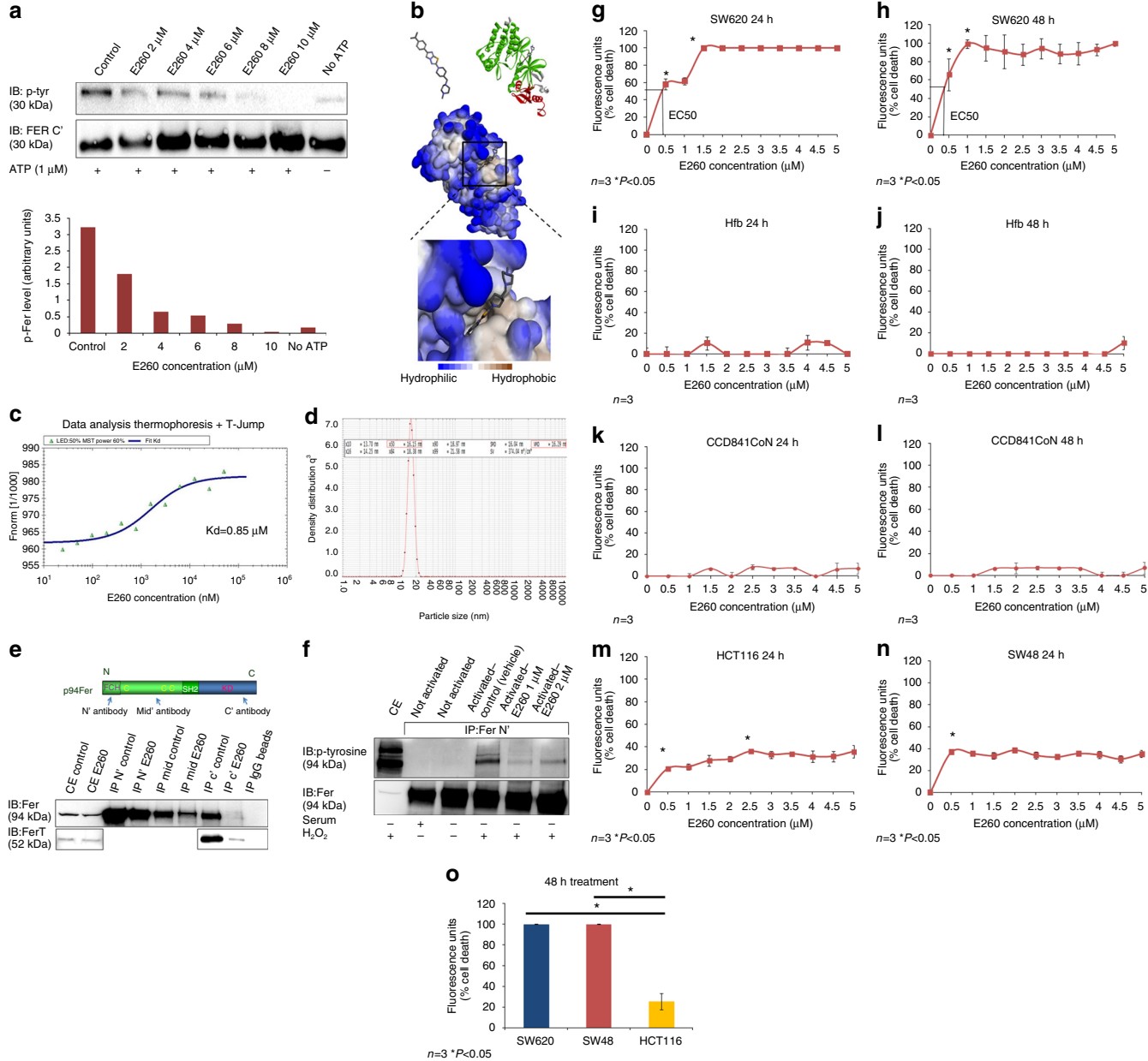

**Fig. 2** E260 binds the Fer/FerT KD and inhibits its activity. **a** In vitro auto-phosphorylation of a recombinant Fer/FerT KD incubated with ATP and ascending concentrations of E260 visualized by WB analysis. Phosphorylation levels (p-tyr) normalized to the Fer/FerT KD levels (Fer C′) are depicted by the histograms presented in the *lower panel*. **b** Computational model of E260 docking into the Fer/FerT KD. **c** MST analysis of E260 binding to Fer/FerT KD and the obtained dissociation constant. **d** Post formulation DLS volume measurement of micelles incorporated with E260. **e** *Upper panel* depicts the Fer protein functional domains and the antibody binding sites used for immunoprecipitation. *Lower panel* presents a WB analysis of the Fer and FerT proteins immunoprecipitated from SW620 cells left untreated or treated with 2 μM E260 for 16 h (*upper blot* and *lower blot*, respectively). One experiment representative of three is presented. **f** IP of Fer from cells in which Fer was not activated or activated (serum starved and $H_2O_2$ treated) and untreated (vehicle alone) or treated with E260 for 12 h. Fer inhibition in cells was manifested by its reduced tyrosine auto-phosphorylation (marked by an *arrow*), which was detected using anti-phosphotyrosine antibodies in a WB analysis. Dose-response curve depicting the death levels of SW620 cells treated with increasing concentrations of E260 for 24 h (**g**) or 48 h (**h**); $n = 3$, ±SE. Dose-response curve depicting the death levels of normal human fibroblast (Hfb) cells treated with increasing concentrations of E260 for 24 h (**i**) or 48 h (**j**); $n = 3$, ±SE. Dose-response curve depicting the death levels of CCD841CoN normal epithelial cells treated with increasing concentrations of E260 for 24 h (**k**) or 48 h (**l**); $n = 3$, ±SE. Dose-response curve depicting the death levels of HCT116 cells (**m**) and SW48 (**n**) cells treated with increasing concentrations of E260 for 24 h; $n = 3$, ±SE. **o** Histograms depicting the death level of SW620 (*blue*), SW48 (*red*), and HCT116 (*yellow*) cells treated with 2 μM E260 for 48 h; $n = 3$, ±SE

yeast cells correlated with their ectopic expression of an active Fer, while inhibition of the kinase activity was expected to relieve this growth attenuation. Advantageously, this assay also selects against Pan assay interference compounds[30], or compounds that cannot cross the cell membrane. The yeast-based HTS system was

implemented in a Tecan multi-functional robotic system, and two automated screening cycles of a library comprised of 100,000 synthetic compounds, led to the identification of the lead 0342 molecule (Fig. 1c, d). 0342 was then subjected to two rounds of structure activity relationship (SAR) analysis leading to the

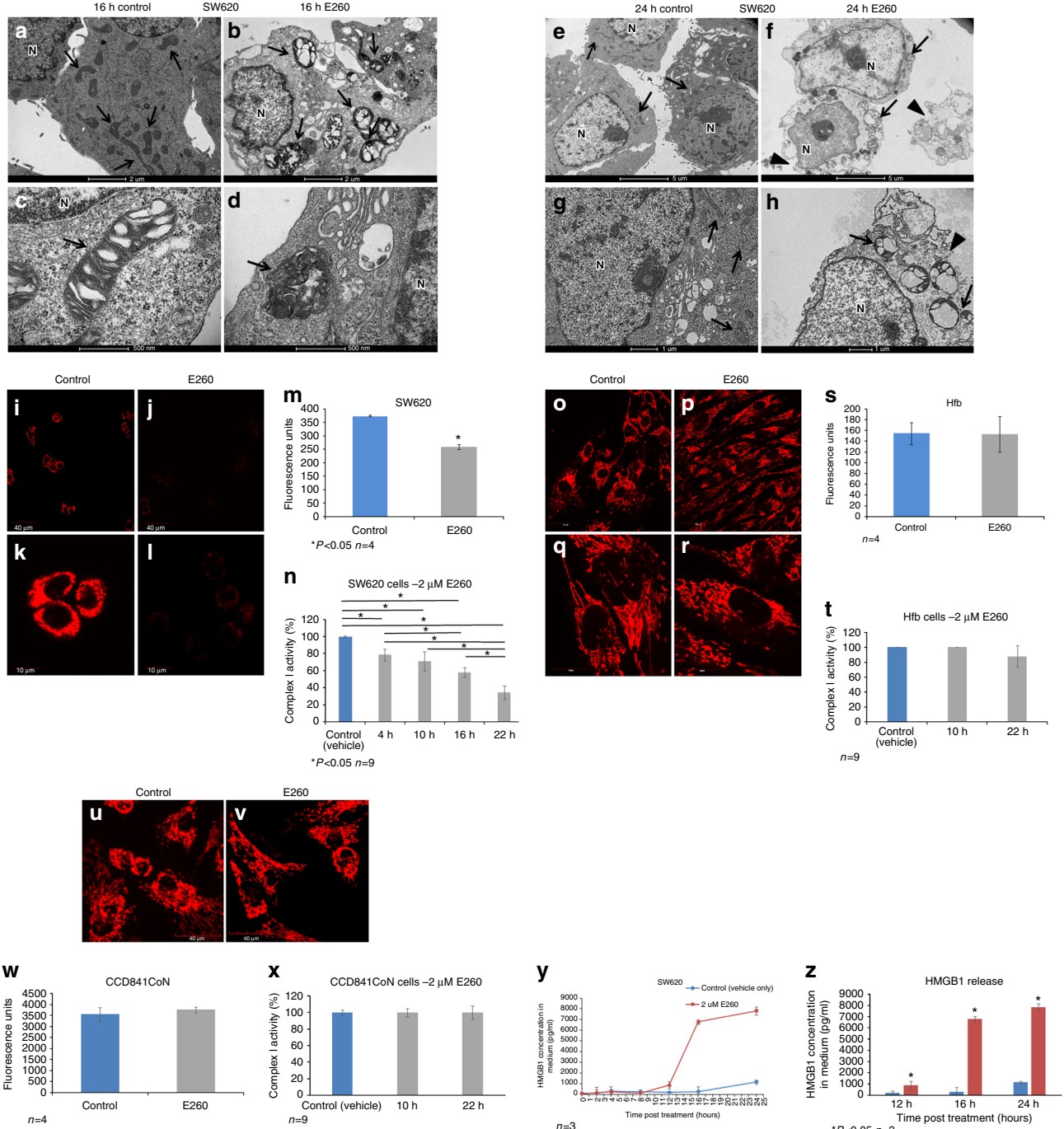

**Fig. 3** E260 induces selective mitochondrial dysfunction and deformation, and necrotic death in CC cells. TEM analysis of SW620 cells treated with vehicle alone for 16 h (**a**, **c**) or 24 h (**e**, **g**), and cells treated with 2 µM E260 for 16 h (**b**, **d**) or 24 h (**f**, **h**). *Arrows* indicate mitochondria. *Arrow heads* point to damaged cell membrane. *Bars* represent 2 µm (**a**, **b**, **e**, **f**), 1 µm (**g**, **h**), and 500 nm (**c**, **d**); N nucleus. MMP visualization by TMRE staining (*red*) in SW620 cells treated with vehicle (**i**, **k**), or with 2 µM E260 (**j**, **l**) for 16 h. **m** ELISA quantification of SW620 MMP in control (*blue histogram*) and in cells treated with 2 µM E260 (*gray histogram*) for 16 h; n = 4, ±SE. **n** Complex I activity in SW620 cells treated with 2 µM E260 for increasing periods of time; n = 9, ±SE. MMP visualization by TMRE staining (*red*) in Hfb cells either treated with the vehicle (**o**, **q**) or with 2 µM E260 (**p**, **r**) for 16 h. **s** ELISA quantification of human fibroblast MMP in control (*blue histogram*) and E260 (*gray histogram*) treated cells; n = 4, ±SE. *Bars* represent 40 µm (**i**, **j**, **o**, **p**) and 10 µm (**k**, **l**, **q**, **r**). **t** Complex I activity in Hfb cells treated with 2 µM E260 for increasing periods of time; n = 9, ±SE. MMP visualization by TMRE staining (*red*) in CCD841CoN cells either treated with the vehicle (**u**) or with 2 µM E260 (**v**) for 16 h. **w** ELISA quantification of CCD841CoN MMP in control (*blue histogram*) and E260 (*gray histogram*) treated cells; n = 4, ±SE. **x** Complex I activity in CCD841CoN cells treated with 2 µM E260 for increasing periods of time; n = 9, ±SE. **y** Quantification of the HMGB1 concentrations in CC cells growth medium following treatment with control solution or 2 µM E260 for different periods of time. n = 3, ±SE. **z** HMGB1 concentrations in CC cells growth medium 12, 16, and 24 h post treatment with control (*blue histograms*) or E260 (*red histograms*) as in **y**, with comparative statistical tests. n = 3±SE

development, synthesis and purification of the E260 inhibitor which was fully characterized by NMR spectra, LC–MS, Fourier transform infrared spectroscopy (FTIR), and high-resolution mass spectra (HRMS) (Fig. 1e, Supplementary Fig. 1C–I). New entity intermediate molecules, which were produced in the synthesis process of E260, were also chemically characterized by NMR, LC–MS, and HRMS analyses (Supplementary Fig. 2). E260 was 50% more efficient than 0342 in restoring the growth of Fer-expressing yeast cells (Fig. 1f, g) and accordingly, it also decreased the tyrosine-phosphorylation level of proteins in the yeast correlating with the MW of the ectopic Fer (Fig. 1h). E260 and 0342 were further tested robotically, and their potential cytotoxic effects on mammalian cells were compared. This analysis demonstrated the selectivity of both 0342 and E260 in compromising the viability of malignant but not normal human cells. Here too, E260 was at least 50% more efficient than 0342 in killing cancer cells (Supplementary Fig. 1A, B).

To demonstrate that E260 directly targets Fer and FerT, an in vitro kinase assay was performed using a purified KD-containing fragment of these enzymes. This analysis demonstrated the direct inhibitory effect of E260 on this domain as reflected by the significantly decreased auto-phosphorylation level of the Fer/FerT KD when incubated with ATP and increasing concentrations of E260 (Fig. 2a). Moreover, computational analysis of E260 docking in the modeled whole Fer protein revealed that the highest scored binding mode of E260 to Fer falls in the ATP-binding pocket of the enzyme's KD (Fig. 2b). To measure the dissociation constant (Kd) of E260 from Fer/FerT KD we performed a microscale thermophoresis (MST) test using ascending concentrations of E260. This analysis corroborated the direct binding of E260 to Fer/FerT KD and determined a Kd of 0.85 µM (Fig. 2c). To optimize the efficacy of E260 in vivo, it was formulated in Cremophor based nano-micelles. The integrity and stability of the obtained micelles were confirmed, and their average diameter was found to range around 16 nm (Fig. 2d).

To examine the effect of the E260 micellar formulation on Fer in malignant cells, the kinase was immunoprecipitated from untreated and from E260-treated SW620 CC cells. While E260 did not disable the immunoprecipitation (IP) of Fer by antibodies directed toward the N-terminal tail of the protein, it did eliminate the immunoprecipitation of Fer and FerT by antibodies directed toward the C-terminal-KD of both Fer forms (Fig. 2e), thereby substantiating the notion that E260 binds the Fer/FerT C-terminal portion. When applied to metastatic grade IV SW620 CC cells, which were serum starved for 16 h and treated with 3 mM $H_2O_2$ to activate Fer[31], E260 exhibited inhibitory effects on the Fer-kinase activity as was reflected by suppressed auto-phosphorylation activity of the enzyme (Fig. 2f). This substantiated the ability of E260 to also inhibit the kinase activity of Fer in cells. To characterize the effect of E260 on malignant cells, metastatic SW620 cells were treated with E260 followed by analysis of viability. Onset of death was observed in the E260-treated cells, with an EC50 value of 400 nM after 24 h of treatment and an EC50 of 300 nM after 48 h (Fig. 2g, h). Importantly, no death was seen in normal human fibroblasts (Hfb) or normal human epithelial cells, that were treated with E260 under the same conditions (Fig. 2i–l) or when subjected to ascending concentrations of E260 under hypoxic conditions (Supplementary Fig. 3A). To further evaluate the selectivity of E260, primary CD34 positive human hematopoietic stem cells were subjected to increasing concentrations of the compound. This analysis also revealed that E260 elicits only marginal effect on these cells viability (Supplementary Fig. 3B). Western blot (WB) analyses also corroborated that similarly to other normal human cells, only Fer but not FerT is expressed in these stem cells

(Supplementary Fig. 3C). In two other CC cell lines, HCT116 and SW48, which are derived from grade II and grade IV colon carcinoma tumors, respectively, the EC50 values for E260 after 24 h of treatment were above 5 µM (Fig. 2m, n). However, while treatment of the grade IV SW48 cells with 2 µM E260 for 48 h led to their complete death, HCT116 cells treated under these conditions reached only 25% cell death (Fig. 2o). The same increased deleterious effect of E260 on metastatic compared to early stage cancer cells was also observed in pancreatic cancer. Accordingly, E260 exhibited an EC50 of 3.2 µM after 72 h treatment of non-metastatic PANC-1 cells, which are derived from a primary pancreatic ductal carcinoma. Moreover, the maximum death level of these cells after 72 h of treatment with E260 was about 70% following treatment with 4 µM E260 (Supplementary Fig. 3D). In comparison, SU.86.86 which are metastatic ductal carcinoma cells, proved to be more susceptible to E260 with an EC50 of 1.1 µM after 72 h of treatment and 100% death level imposed by 2 µM E260 (Supplementary Fig. 3E).

**E260 selectively evokes necrotic death in CC cells**. To characterize the death process imposed by E260 in the treated malignant cells, a transmission electron microscopy (TEM) analysis was performed on SW620 cells treated with E260. This revealed the deformation and loss of the normal elongated shape of mitochondria in cells treated with E260 for 16 h (Fig. 3a–d). Closer examination revealed that the cristae of the damaged mitochondria collapsed and the mitochondria swelled (Fig. 3d). After 24 h of treatment, the mitochondria underwent lysis, and, while integrity of the nuclei was preserved, disruption of the cytoplasmic membrane of the treated cells could be seen (Fig. 3e–h). Disintegration of the plasma membrane accompanied by preserved integrity of the cell nucleus indicated the onset of necrotic rather than apoptotic death in the E260-treated cells[32]. To corroborate the lack of apoptosis in E260-induced cell death, the absence of apoptotic markers including cleaved caspase 3, cleaved PARP-1 and translocation of AIF to the nucleus[18, 33] was confirmed (Supplementary Fig. 4A–D). In addition, Annexin V–propidium iodide (PI) analyses did not indicate a significant increase in early apoptotic cell population which is stained only by Annexin V, but rather depicted the profound positive PI/Annexin V staining which is correlated with the onset of necrotic death in the treated cells[34] (Supplementary Fig. 4E–F). In addition, uptake of EthD III, which is a hallmark of necrotic death, was also seen in the E260-treated cells (Supplementary Fig. 5A–D)[35]. Finally, co-treatment of CC cells with E260 and either the apoptosis inhibitor-Z-VAD[36], the ferrpotosis inhibitor—Ferrstatin-1(Fer-1)[37], or the necroptosis inhibitor—Necrostatin-1 (Nec-1)[38], indicated that necroptosis takes part in the cellular death inflicted by E260, as seen by the attenuation of the E260 cytotoxic effect exerted by the combined treatment with Nec-1[39] (Supplementary Fig. 5E–G). To follow the kinetics of necrosis onset in E260-treated SW620 cells, the release of the necrotic marker—high-mobility group protein 1 (HMGB1) to the cells growing medium[8], was determined. Release of HMGB1 from the treated cells was detected as early as 12 h post treatment with a significant increase after 16 and 24 h (Fig. 3y, z). The results of these assays were consistent with the TEM analysis, indicating that necrotic rather than apoptotic death is evoked by E260 in CC cells.

Destruction of the mitochondria was accompanied by depolarization of the mitochondrial membrane potential (MMP) in E260-treated malignant cells (Fig. 3i–m). Notably, the activity of the ETC complex I was significantly decreased as early as 4 h following subjection of CC cells to E260 and this

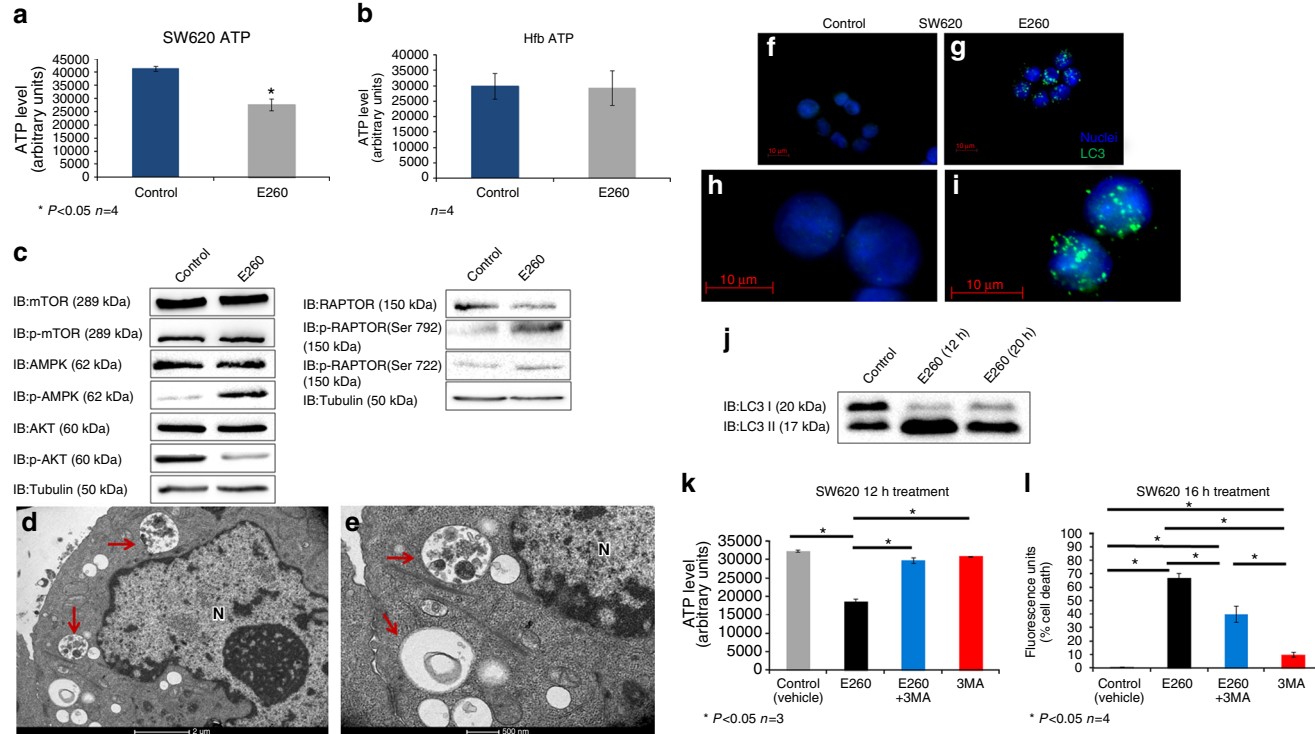

**Fig. 4** Decreased ATP level and induction of autophagy in E260-treated SW620 cells. ATP levels in SW620 **a** and Hfb **b** cells treated with vehicle (*blue histogram*), or 2 μM E260 (*gray histogram*) for 12 h; $n = 4$, ±SE. **c** Protein lysates from SW620 cells treated with control or 2 μM E260 for 12 h were subjected to WB analysis of the total expression or phosphorylation level (p-X) of selected proteins from the AMPK–mTOR signaling pathway. **d**, **e** TEM analysis of SW620 cells treated with 2 μM E260 for 12 h. *Red arrows* indicate autophagosomes. N nucleus. *Bars* represent 2 μm in **d** and 500 nm in **e**. Immunostaining for LC3 (*green*) of SW620 cells treated with vehicles (**f**, **h**) or with 2 μM E260 (**g**, **i**) for 12 h. Nuclei were visualized with Hoechst (*blue*). *Bars* represent 10 μm. **j** Protein lysates from SW620 cells treated with control or 2 μM E260 for 12 or 20 h were subjected to WB analysis using anti-LC3 antibody. **k** ATP levels in SW620 cells treated for 12 h with vehicle (*gray*), 2 μM E260 (*black*), 2 μM E260, and 5 mM 3-MA (*blue*), and 5 mM 3-MA alone (*red*); $n = 3$, ±SE. **l** Cell death levels in SW620 cells treated for 16 h with vehicle (*gray*), 2 μM E260 (*black*), 2 μM E260, and 5 mM 3-MA (*blue*), or 5 mM 3-MA alone (*red*); $n = 4$, ±SE

decrease was progressively exacerbated over time (Fig. 3n). Remarkably, these effects were not seen in E260-treated normal human fibroblasts or epithelial cells, which demonstrated unaffected and normal MMP and complex I activity (Fig. 3o–x).

**E260 induces ATP decrease and autophagy in CC cells**. Deformation of mitochondria and reduced MMP suggest a disruption of the energy balance in the treated CC cells. To support this finding, the cellular ATP content was determined in untreated and in E260-treated CC cells. A decrease of 35%–40% was observed in the cellular ATP level of CC cells treated with E260 for 12 h, a time point preceding the onset of death (Fig. 4a). Notably, this effect of E260 on the cellular ATP level was not seen in Hfb after 12 h of treatment (Fig. 4b). Mitochondrial damage and decrease in the cellular ATP level could direct the initiation of autophagy in the E260-treated CC cells[40]. In accordance with the observed energy deficit, the metabolic sensor AMPK was phosphorylated and activated[40, 41] in the E260-treated cells. This was accompanied by the inhibitory phosphorylation of the AMPK down-stream effector-Raptor on serines 792 and 722 and by a decrease in the activating phosphorylation of Akt (Fig. 4c)[41]. Phosphorylation of Raptor leads to its dissociation from the mTOR complex, thereby resulting in inactivation of mTOR kinase activity regardless of its phosphorylation state, which indeed was not changed in the E260-treated cells (Fig. 4c)[41, 42]. Deactivation of mTOR relieves its inhibitory effect on autophagy, thereby enabling the activation of this process[40, 42]. In accordance with this notion, autophagy was induced and autophagosomes

accumulated in the treated CC cells, as was observed by TEM analysis (Fig. 4d, e) and immunostaining for the autophagosome marker—LC3 (Fig. 4f–i)[43]. Onset of autophagy was also evident from the relative increase in the level of the autophagosome-associated LC3 form—LC3II, in the E260-treated cells (Fig. 4j)[43]. To examine whether the induced autophagy acts as a salvage process, or whether it contributes to the evoked death, we simultaneously treated SW620 cells with both E260 and the autophagy inhibitor-3-methyladenine (3-MA). Surprisingly, inhibition of the E260-induced autophagy with 3-MA almost completely restored the ATP level in the E260-treated malignant cells (Fig. 4k). Accordingly, inhibition of the induced autophagy reduced the level of death, although not completely restoring the viability of the cells treated with E260 for up to 16 h (Fig. 4l). 3-MA seemed to postpone the onset of cell death evoked by E260, since cells co-treated with E260 and 3-MA reached 100% death only after 96 h (Supplementary Fig. 6). This suggests that the induced and persisting autophagy consumes ATP and contributes to the necrotic death imposed by E260.

**E260 induces PARP-1 activity which promotes cell death**. The incomplete restoration of cell viability upon inhibition of autophagy, suggested the existence of additional cellular pathway (s) that contributes to the onset of death in the E260-treated CC cells. To identify cellular pathways in which Fer is engaged in malignant cells, and proteins that interact with this kinase, a proteomic, Fer-interactome analysis was carried out. One of the proteins that was repeatedly identified by this analysis was

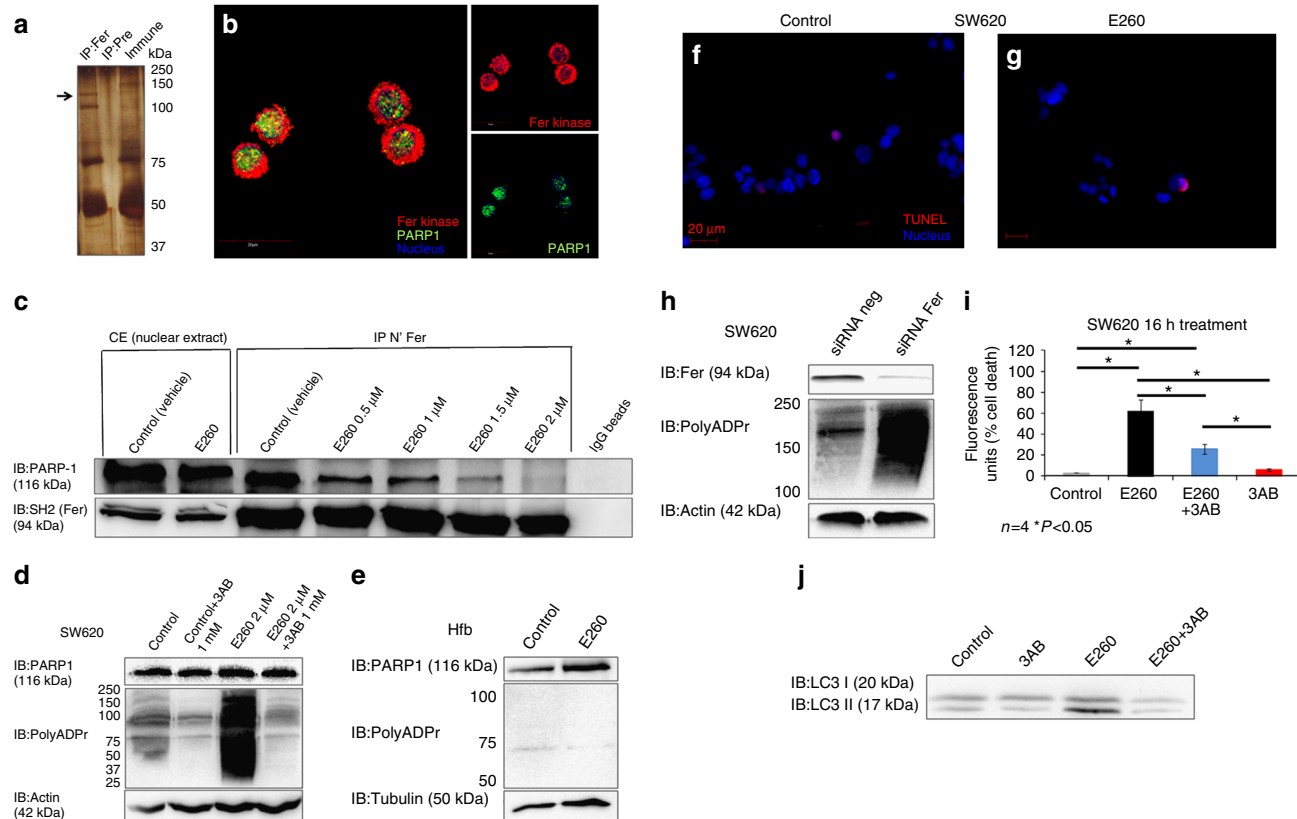

**Fig. 5** E260 dissociates Fer from PARP-1 leading to PARP-1 activation in CC cells. **a** Fer was immunoprecipitated from SW620 cells using anti-Fer antibodies or pre-immune serum, which served as a control. Co-immunoprecipitated proteins were resolved by SDS–PAGE, stained with Silver-stain kit, excised from the gel, and sent for MS analysis. **b** Co-immunostaining of both Fer (*red*) and PARP-1 (*green*) in SW620 cells. Co-localization of the two proteins is indicated in *yellow*. Nuclei were stained with Hoechst (*blue*). *Bar* represents 20 μm. **c** Nuclear proteins lysates from SW620 cells treated with control (vehicle solution) or increasing concentrations of E260 for 12 h were subjected to IP of Fer using antibodies directed toward the N-terminal tail of the kinase (N' Fer) and analyzed in WB using specific antibodies. IgG-linked beads served as a control. **d** WB analysis of lysates prepared from control treated or SW620 cells treated with E260 for 12 h. Lysates were reacted with anti-PARP-1, anti-Poly-ADPr, and anti-actin, antibodies. **e** WB analysis of protein lysates prepared from control or E260-treated Hfb cells using anti-PARP-1, anti-Poly-ADPr, and anti-tubulin, antibodies. SW620 cells were treated with control solution **f**, or E260 **g** for 12 h followed by a TUNEL assay to detect DNA damage (*red*). Nuclei were stained with Hoechst (*blue*). *Bars* represent 20 μm. **h** WB analysis of protein lysates prepared from SW620 cells treated with non-relevant siRNA (*siRNA neg*) or siRNA directed toward the *fer* mRNA (*siRNA-Fer*). Levels of Fer, Poly-ADPr and actin were determined using specific antibodies. **i** Cell death in SW620 cells treated with vehicle (*gray*), E260 for 16 h (*black*), E260, and the PARP-1 inhibitor 3-AB (*blue*), or with 3-AB alone (*red*); $n = 4$, ±SE. **j** Protein lysates from SW620 cells treated for 12 h with vehicle, 3-AB, E260, or E260 + 3AB were subjected to WB analysis using anti-LC3 antibodies. All images shown are representative of five independent experiments, which gave similar results

PARP-1 (Fig. 5a). Immunostaining for both Fer and PARP-1 revealed the co-localization of the two proteins in the cell nucleus (Fig. 5b). Co-immunoprecipitation analysis confirmed the association of Fer with PARP-1 in nuclear lysates prepared from CC SW620 cells (Fig. 5c). Notably, treatment with increasing concentrations of E260 gradually dissociated Fer from PARP-1, and the two proteins failed to co-immunoprecipitate from lysates of CC cells treated with 2 μM E260 (Fig. 5c). After corroborating the dissociating effect of E260 on the Fer-PARP-1 complex, we examined whether E260 also affects PARP-1 activity. Strikingly, CC (Fig. 5d) but not normal cells (Fig. 5e), exhibited a significant increase in the poly-ADP-ribosylation activity of PARP-1, when subjected to E260. This dysregulated activity was abolished upon exposure of the E260-treated malignant cells to the PARP-1 inhibitor, 3-aminobenzamide (3AB)[44] (Fig. 5d), confirming the direct involvement of PARP-1 in this induced poly-ADP-ribosylation activity.

To exclude the possibility that DNA damage induced by E260 is responsible for the activation of PARP-1[45], we treated SW620 cells with E260 for 12 h, a time period that precedes cell death,

and during which PARP-1 becomes activated. Cells were then subjected to terminal deoxynucleotidyl transferase (TUNEL) analysis[46], which did not reveal any effect of E260 on DNA integrity in the treated CC cells (Fig. 5f, g). The E260 directed dissociation of Fer from PARP-1 and the accompanying induction of the PARP-1 activity suggested involvement of Fer in the regulation of the PARP-1 function. This was substantiated by a siRNA directed Fer knockdown experiment, which also led to an increase in the PARP-1 ADP-poly-ribosylation activity (Fig. 5h). Upregulated activity of PARP-1 consumes ATP and can drive necrotic death in mammalian cells[28, 47]. To directly examine the involvement of the increased PARP-1 activity in the cellular death evoked by E260, CC cells were simultaneously subjected to E260 and to the PARP-1 inhibitor, 3AB. The presence of 3AB significantly decreased the death induced by E260 in CC cells by 50% (Fig. 5i). This supports the notion that elevated PARP-1 activity contributes to the cytotoxic effect of E260 in CC cells. PARP-1 can act as a positive regulator of autophagy[27, 47], and notably, inhibition of the PARP-1 activity decreased the level of induced autophagy in the E260-treated

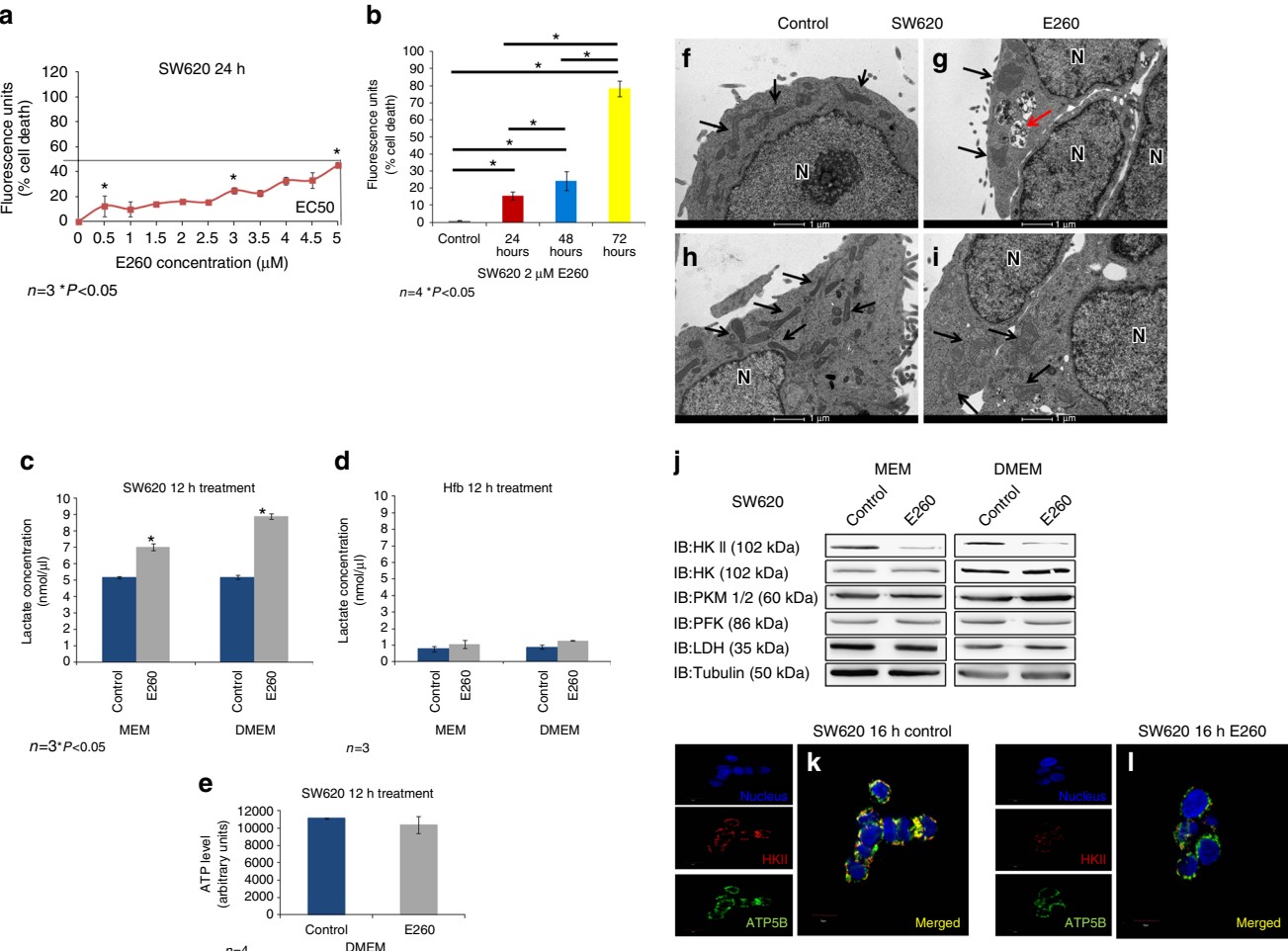

**Fig. 6** Upregulated glycolysis and delayed onset of death in SW620 cells treated with E260 under high glucose conditions. **a** Death of SW620 cells grown in DMEM and treated with increasing concentrations of E260; $n = 3$, ±SE. **b** Death of SW620 cells grown in DMEM and treated with 2 μM E260 for different periods of time; $n = 4$, ±SE. **c** Lactate levels secreted by SW620 cells grown either in MEM or DMEM and treated with control vehicle (*blue*) or 2 μM E260 (*gray*) for 16 h. **d** Lactate levels secreted by Hfb grown either in MEM or DMEM and treated with control vehicle (*blue*) or 2 μM E260 (*gray*) for 16 h. **e** ATP content in SW620 cells grown in DMEM and treated with control (*blue*) or 2 μM E260 (*gray*) for 12 h; $n = 4$, ±SE. **f–i** TEM analysis of SW620 cells grown in DMEM and treated with control vehicle **f**, **h**, or 2 μM E260 **g**, **i** for 24 h. *Black arrows* depict mitochondria. *Red arrow* points to autophagosomes. *Bars* represent 1 μm. N nucleus. **j** Protein lysates prepared from SW620 cells grown in MEM or DMEM and treated with control vehicle or E260 for 16 h, were subjected to WB analysis with antibodies directed toward selected glycolytic enzymes. SW620 cells treated with control vehicle (**k**) or 2 μM E260 (**l**) for 16 h were subjected to immunostaining using anti-HKII (*red*), and anti-ATP5B (*green*) antibodies. Nuclei were visualized with Hoechst (*blue*). *Bars* represent 10 μm. Separated channels are presented to the *left* of the merged images. *Yellow color* indicates co-localization

cells, as reflected by the decreased ratio of LC3II to LC3 I upon inhibition of PARP-1 (Fig. 5j). These findings support the roles of PARP-1 in the induction of autophagy and cell death in E260-treated CC cells.

**High glucose delays the death of E260-treated CC cells.** In parallel to their mitochondrial energy production, cancer cells efficiently utilize glycolysis for both energy production and anabolic metabolism[3]. We therefore studied the effect of extracellular glucose concentrations on the E260 cytotoxicity in SW620 cells. While in the presence of 1 g/l extracellular glucose, 2 μM E260 inflicted 100% cell death of CC cells within 24 h, with an EC50 of 400 nM (Fig. 2g), under 4.5 g/l glucose (DMEM), SW620 CC cells treated with E260 demonstrated only 50% death at the highest dose with an EC50 of 5.1 μM (Fig. 6a). Nevertheless, it should be noted that 100% cell death was still observed when cells grown under high glucose concentration were treated with 2 μM E260 for 72 h (Fig. 6b).

To examine whether the delayed death of CC cells treated with E260 under high glucose is linked to upregulated glycolysis, we compared the glycolytic rate in untreated cells versus that in cells treated with E260 for 12 h by measuring the concentrations of secreted lactate[48]. This analysis revealed that CC cells accelerated their glycolytic rate by 35% when subjected to E260 (Fig. 6c). The rate of glycolysis was further accelerated by an additional 30% under high glucose concentration resulting in a 65% elevation in the glycolytic rate in E260-treated cells (Fig. 6c). The significant elevation in the glycolytic rate of E260-treated CC cells was not seen in Hfb cells treated with E260 under low or high glucose (Fig. 6d), further demonstrating the selective effects of E260 on malignant cells. To support the notion that upregulated glycolysis under high glucose conditions transiently opposes the effect of E260 on the ATP level in CC cells, we quantified the cellular ATP level in untreated and in E260-treated CC cells. This analysis confirmed that high glucose medium interferes with E260 activity, reducing its effect on ATP levels in treated CC cells (Fig. 6e). Although death of CC cells subjected to E260 and grown under

high glucose conditions was delayed, mitochondrial damage was still observed when these cells were treated with E260 for 24 h and examined by TEM (Fig. 6f–i). The inability of enhanced

glycolysis to salvage the cells from cellular death might be linked to the fact that the mitochondrial damage caused by E260 was accompanied by downregulation of HK II (Fig. 6j), which

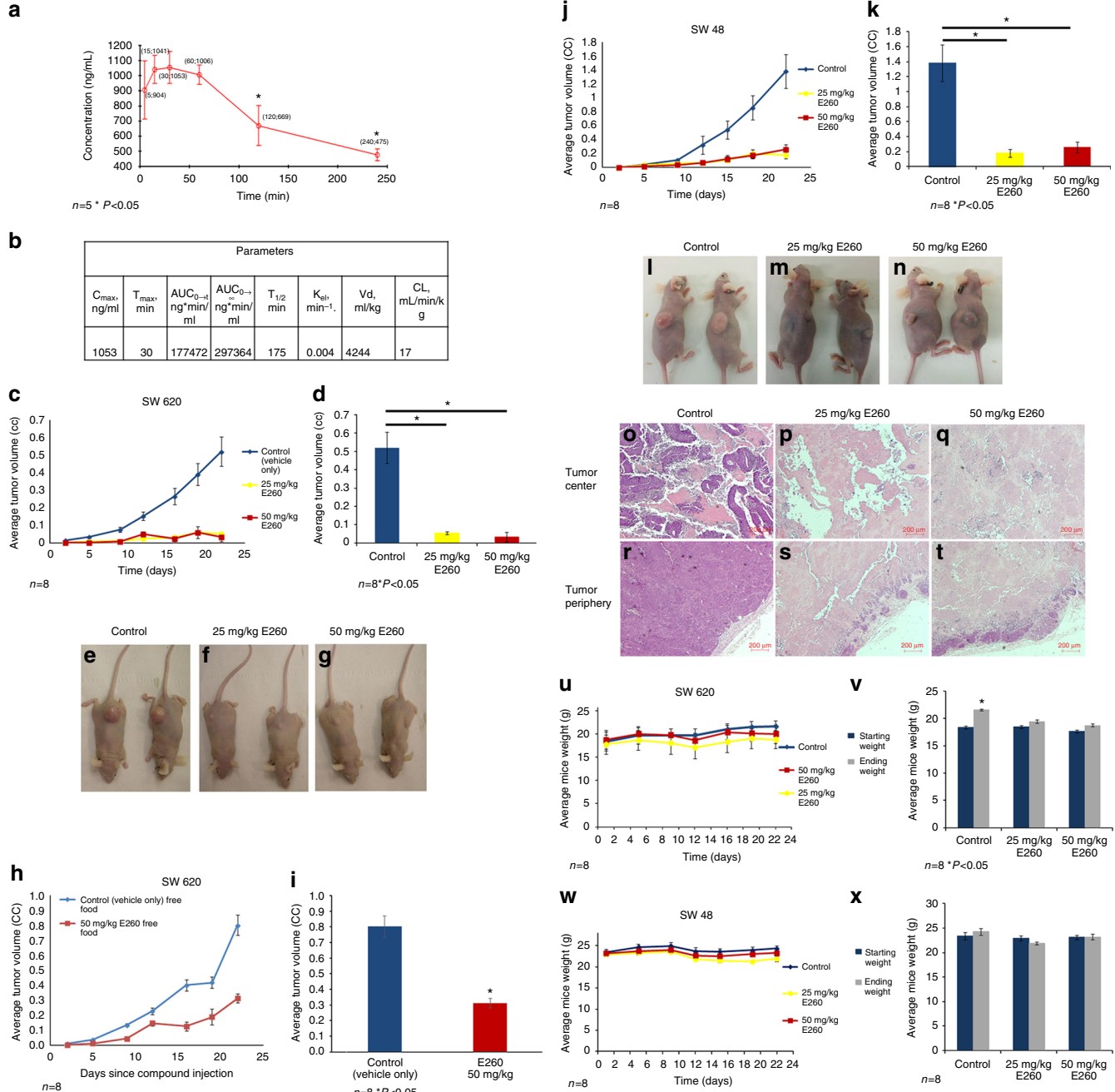

**Fig. 7** Systemic E260 treatment attenuates CC xenograft progression in immuno-compromised mice. **a** PK analysis of E260 in mice blood following IP administration of the compound; $n = 5$ for each time point. **b** Summary table for E260 PK parameters and distribution in vivo. **c** Tumor progression in mice bearing SW620 derived xenografts fed with restricted diet and injected IP twice a day with control vehicle, 25 mg/kg E260, or 50 mg/kg E260; $n = 8$, ±SE. **d** Average tumor volume after 22 days of treatment: control (*blue*), 25 mg/kg E260 (*yellow*), and 50 mg/kg E260 (*red*); $n = 8$, ±SE. **e–g** Representative images of mice bearing SW620 xenografts at day 22 post injection. **h** Tumor progression in mice bearing SW620 derived xenografts fed with ad libitum diet and injected IP twice a day with vehicle or 50 mg/kg E260; $n = 8$, ±SE. **i** Average tumor volume after 22 days of treatment with vehicle (*blue*) or 50 mg/kg E260 (*red*); $n = 8$, ±SE. **j** progression in mice bearing SW48 xenografts and injected IP twice a day with vehicle, 25 mg/kg E260, or 50 mg/kg E260; $n = 8$, ±SE. **k** Average tumors volume after 22 days of treatment: control (*blue*), 25 mg/kg E260 (*yellow*) and 50 mg/kg E260 (*red*); $n = 8$, ±SE. **l–n** Representative images of mice bearing SW48 xenografts at day 22 post injection. **o–t** Representative images of tumor sections dissected from mice at day 22 of treatment with control (vehicle only) (**o**, **r**), 25 mg/kg E260 (**p**, **s**), or 50 mg/kg E260 (**q**, **t**). *Bars* represent 200 μm. **u**, **w** Weight profiles of mice bearing SW620 (**u**) or SW48 (**w**) xenografts and IP injected twice a day with control, 25 mg/kg E260, or 50 mg/kg E260; $n = 8$, ±SE. **v**, **x** Average starting weights and end-point weights of mice bearing tumors derived from SW620 (**v**) or SW48 (**x**) cells after 22 days of twice daily IP injection: control, 50 mg/kg E260, or 25 mg/kg E260; $n = 8$, ±SE

associates with the mitochondrial outer membrane and serves as the key initiating enzyme of the glycolytic pathway in cancer cells[4]. Interestingly, neither the level of HKI, which is the main glycolysis-initiating enzyme in normal cells[49], nor that of other glycolytic enzymes was affected under E260 treatment (Fig. 6j). The downregulation of HK II was also demonstrated by its decreased presence on the mitochondria of E260-treated cells (Fig. 6k, l).

**E260 suppresses xenografts progression in vivo.** After characterizing the effect of E260 on cancer cells in vitro, we turned to study its effect in vivo. As an initial step, we determined the pharmacokinetic (PK) profile of E260 in mice. E260 exhibited a T1/2 of 175 min in the blood, and a volume of distribution of 4244 ml/kg suggesting an efficient distribution of the compound in the animal tissues[50] (Fig. 7a, b). Based on our in vitro results, we aimed to maximize the efficacy of E260 in mice xenografts model by maintaining the animals on a restricted diet, which limits their glucose consumption (see Methods section). This diet lowered the blood glucose level by about 35% in the treated and untreated animals (Supplementary Fig. 7A). To evaluate the efficacy of E260 on tumor growth, SW620 cells were subcutaneously introduced into immuno-compromised "Nude" mice. Administration of E260 led to a significant attenuation of tumor progression throughout the experiment, and to a 10-fold decrease in average tumor volume after 22 days of treatment (Fig. 7c–g). When the tumor bearing mice were allowed an ad libitum food consumption E260 attenuated the CC tumor progression by 2.5-fold only (Fig. 7h, i) correlating with the in vitro attenuating effect of high glucose on the E260 cytotoxic activity (Fig. 6a, b). To further demonstrate the anti-cancer activity of E260 in vivo, mice bearing SW48 cells derived xenografts were treated with the compound and the tumor progression profiles were determined. Mice treated with E260 demonstrated a 5-6-fold attenuation in tumors progression when compared to the control treated group (Fig. 7j–n). Histopathological examination of the SW620 derived tumors dissected at day 22 revealed a highly vascularized and viable tissue at the center and periphery of tumors removed from control treated mice (Fig. 7o, r). In contrast, tumors dissected from the E260-treated mice were comprised mostly of dead necrotic and un-vascularized tissue, both at the tumor center and periphery (Fig. 7p–q, s–t). Importantly, no significant effect on the weight of the treated animal's was observed in the in vivo experiments (Fig. 7u–x). Accordingly, the daily injection of E260 to these tumors bearing mice did not cause any toxic or adverse side effects as determined by measuring blood values of markers of liver and kidney function and blood electrolytes (Supplementary Fig. 7B–D and Supplementary Fig. 8A–F). Histopathological analyses also failed to detect any abnormalities that might be related to toxicity in the major organs including the heart (Supplementary Fig. 9A–L), kidneys (Supplementary Fig. 10A–L), and liver (Supplementary Fig. 11A–L).

## Discussion

Reprogrammed glycolytic and mitochondrial pathways are hallmarks of the altered energy generation system of malignant cells, which support the abnormal survival and proliferation needs of these cells[51, 52]. These unique metabolic pathways can be therefore exploited as selective targets for cancer intervention[52]. Moreover, recent studies suggest that when cancer cells progress toward aggressive and metastatic stages, their dependence on functional reprogrammed mitochondria becomes more prominent, thereby making the mitochondria an attractive target for cancer therapy[5, 53].

Fer and its truncated sperm and cancer cell-specific variant, FerT, are newly discovered components of the mitochondrial reprogrammed ETC of cancer cells[7]. In this study, we describe the development of a novel Fer and FerT inhibitor named E260, which inhibits phosphorylation activity of these kinases and exerts potent and selective anti-neoplastic effects. When incorporated into nanoscale micelles, E260 exhibited an EC50 of 400 nM and imposed a complete death of metastatic CC cells after 24 h of treatment. Interestingly, the high-grade metastatic CC and pancreatic cancer cells were more sensitive to E260 than low grade CC and pancreatic cancer cells, and no death was observed in treated normal human cells. Hence, while kinase inhibitors such as Staurosporine often exert broad range off-target inhibition, thereby demonstrating a non-selective cytotoxicity which eliminates both cancer and normal somatic cells[54, 55], E260 is a more selective agent. The selective death induced by E260 in malignant cells might be attributed to the specific association of Fer and FerT with the ETC complex I of cancer but not of normal somatic cells[7]. Accordingly, a selective inhibition of complex I was seen in the treated CC cells but not in normal cells. This coincides with the observed decrease in complex I activity in cells subjected to Fer-targeting or FerT-targeting siRNAs[7], thereby supporting the inhibitory effect of E260 on the Fer and FerT activities in cells. Notably, mitochondria of E260-treated cancer cells underwent deformation and decrease in MMP, an effect that was not seen in normal cells. These imposed mitochondrial abnormalities are similar to the deleterious effects exerted by other ETC inhibitors such as rotenone, TTFA and chlorimipramine, which inhibit ETC complex I, II, and III activity, respectively[56–58]. Furthermore, cancer cells bearing severe mitochondrial respiration defects also exhibit deformed and swollen mitochondria, emphasizing the fact that mitochondrial integrity is compromised when the ETC function is impaired[59]. Concomitant with the mitochondrial deformation, E260-treated CC cells exhibited a decrease in ATP content of up to 50%, and this decrease did not occur in treated normal Hfb cells, further indicating the selectivity of E260. Low production of mitochondrial ATP can direct the onset of necrotic death[32]. However, cancer cells can maintain a balanced and compensating energy generation system by accelerating their glycolysis to a much higher rate than that occurring in normal somatic cells[3, 59]. Indeed, CC cells treated with E260 exhibited an elevated glycolytic rate in their attempt to potentiate ATP production. This trend was further accelerated when the treated CC cells were allowed to grow under high glucose concentration, alleviating the decrease in ATP level, and delaying the onset of cell death.

Interestingly, E260 did not cause a significant elevation in the glycolytic rate in normal human cells, reinforcing the fact that these cells do not need to compensate for ATP loss. Although the enhanced glycolysis delayed the E260-imposed death in CC cells, it did not prevent it. The limited effects of upregulated glycolysis might be linked to the downregulation of HK II, which accompanies the deformation of mitochondria by E260. Thus, E260 cooperatively impairs the functioning of two key energy generation processes in malignant cells, affecting both the mitochondrial and the glycolytic systems. Cells experiencing energy depletion and/or organelle damage can utilize the autophagy processes to maintain homeostasis and escape cellular death[25]. A specific example of such a pathway is the complex I non-selective inhibitor, rotenone, which induces mitochondrial dysfunction leading to the activation of autophagy[56]. Similarly, selective complex I inhibition and onset of autophagy are evoked in E260-treated CC cells. E260 directed ATP deficiency led to the activation of AMPK, deactivation of mTOR and activation of autophagy[25, 41, 60]. However, despite of the induction of

autophagy, almost all the mitochondria in the E260-treated CC cells were deformed and functionally impaired. This suggests the inability of the induced autophagy to recycle the severely damaged mitochondria and to salvage the energetically deprived cells. Moreover, the persistent activation of autophagy consumes ATP, thereby contributing to the energy crisis and onset of necrotic death in the E260-treated cells. By applying the necroptosis inhibitor-Nec-1[38] we further demonstrated the involvement of this programmed necrotic process in the death evoked by E260 in malignant cells[39]. The E260-induced autophagy and consequent necrosis seems also to be driven by a deregulated PARP-1, which is dissociated by E260 from its newly discovered regulator, Fer. Moreover, by excluding the possibility that E260 induces DNA damage, which would also activate PARP-1, we substantiate the notion that E260 leads to the activation of PARP-1 through its dissociation from Fer. PARP-1 supports autophagy through the AMPK–mTOR pathway, and its activity or knockdown can promote or prevent stress mediated autophagy, respectively[27]. We suggest that by acting simultaneously through at least two subcellular targets, mitochondria and PARP-1, of malignant but not of normal cells, E260 selectively imposes ATP deficiency, which is exacerbated by a continuously active process of autophagy. Although in this study we used 3AB as a PARP-1 inhibitor we cannot rule out the fact that 3AB may also inhibit other members in the PARP family, which could also contribute to the cellular effects shown in the current study. These combined and selective effects of E260 lead to the necrotic death of malignant cells without affecting normal cells. Importantly the potent and selective anti-cancer effect of E260 was also manifested in vivo.

Collectively, we describe here the development of a novel low molecular weight synthetic compound, E260, which targets Fer and FerT. By cooperatively activating energy-consuming processes and selective mitochondrial deformation in malignant cells, E260 is selectively lethal to cancer cells both in vitro and in vivo.

## Methods

**Plasmid construction**. DNA fragments encoding native, HA-tagged Fer (Ha-Fer), and Ha-Fer bearing an inactivating mutation at the 715 position (Y715F) were cleaved from the previously constructed expression—plasmids PECE HA-Fer and PECE HA-Fer[Y715F23], respectively, and were ligated to the pAES426-expression vector (pAES). The Fer/FerT Kinase Domain (KD) (541–822 a.a) encoding fragment was cloned into the pRSFDuet-1 expression plasmid, and transformed into *Escherichia coli* BL-21, strain[23].

**Primers**. The following primers were used for Fer KD: Forward-5′-GCCGCGAAT TCGAAATCAGGTGTAGTTCTGCT-3′ Backward- 5′-CGACTGCGGCCGCCTA TGTGAGTTTTCTCTTGAT-3′

**Tissue culture**. The following cell lines were used: Colon cancer cell lines: HCT116, HT29, RKO, SW620, SW48. Liver cancer cell line: Hep3B. Normal human cells: Hfb (human fibroblasts from human foreskin), CCD841CoN (Normal human epithelial cells) and CD34 positive hematopoietic donor derived stem cells. Pancreatic cancer cells: PANC-1, SU.86.86. All cell lines were obtained from the American Type Culture Collection and grown in minimal essential medium (MEM), as described previously[7], unless stated otherwise.

Cells were authenticated using a morphology check by microscope and by characterization of the DNA profiles using short tandem repeat analysis.

**Hypoxia induction**. Cells were grown for 24 h in MEM (Biological Industries) at 37°C with 5% CO2, and were then transferred for additional 24 h to anaerobic culture jar containing CO2-generating envelope (GasPak EZ, BD). These conditions reduce the oxygen level in the jar to 1% within 30 mi.

**Yeast growth and transformation**. *Saccharomyces cerevisiae* BY4741-3702 (3702) strain cells bearing a deletion in the *ptc1* gene, which encodes the PP3C phosphatase were grown in normal medium. Transformation of the various expression plasmids to the 3702 yeast cells and yeast growth post-transformation were carried out as previously described[29]. Yeast growth analyses were carried out by

inoculating the cells into a 96-well plate with a starting optical density (OD) of 0.005 in each well. Each well was then treated with either dimethyl sulfoxide (DMSO) (Sigma-Aldrich, Israel), 0342 dissolved in DMSO, or E260 dissolved in DMSO. The plates were shaken constantly and incubated for 60 h at 30 °C. OD in each well was determined at seven time points during incubation, at intervals of 10 h.

**Western blot analysis**. Whole-cell lysates were prepared as follows: cells were collected and were then lysed with Tris buffer pH 7.5 lysis buffer containing: 20 mM Tris-HCl, 1 mM EDTA, 150 mM NaCl, 2 mM sodium orthovanadate, 1% NP40, 1 mM NaF (sodium fluoride), 10 mM β-glycerophosphate, and supplemented with complete protease inhibitors cocktail (Complete mini Sigma 0469312400 ROCHE). In total, 30 µg protein lysates from each sample were resolved by 7.5% or 10% SDS–PAGE, and analyzed by western blotting. Protein bands were visualized using a horseradish peroxidase (HRP)-conjugated secondary antibody to rabbit or mouse IgG (Jackson) in conjunction with a WB chemiluminescence reagent (Pierce cat. 34080).

All full length and un-cropped blots which correspond to the WB images that appear in this manuscript are available at Supplementary Fig. 12 and Supplementary Fig. 13.

**Antibodies**. The following primary antibodies were used in various analyses: anti-HA tag (Roche 11583816001, USA, 1:1000), anti-actin (Santa-Cruz sc-1616-R, USA, 1:1000), anti-LC3 (by Sigma-Aldrich L7543, USA, 1:1000), anti-HKI and anti-HKII (Cell signaling mAB2024, mAB2867, USA, 1:1000), anti-pyruvate kinase isozymes M1/M2 (Cell Signaling mAB3190, USA, 1:1000), anti-phosphofructokinase-1 (Cell Signaling mAB8164 1:500), anti-lactate dehydrogenase (Cell Signaling mAB3582, USA, 1:1000), anti-Tubulin (Abcam ab7750, USA, 1:1000), anti-poly [ADP-ribose] polymerase 1 (PARP-1) (Santa-Cruz sc-8007, USA, 1:1000), anti-poly-ADP-ribosylation (PolyADPr) (Alexis Alx-804220, USA, 1:1000), anti-Fer (Sigma-Aldrich HPA007641, USA, 1:500), anti-Fer SH2 (produced by our laboratory, 1:500), anti-Fer N′ terminus (produced by our laboratory, 1:500), anti-Fer C′ terminus (produced by our laboratory, 1:500), anti-p-AMPK (Cell Signaling 4188, USA, 1:500), and anti-phosphotyrosine (anti-pY-Exalpha X-1021, USA, 1:500), anti-mTOR (Cell Signaling mAB2983, USA, 1:100), anti-p-mTOR (Cell Signaling mAB5536, USA, 1:1000), anti-Akt1/2/3 (Santa-Cruz sc-8312, USA,1:1000), anti-p-Akt1/2/3 (Santa-Cruz sc-7985-R, USA, 1:1000), anti-Regulatory-associated protein of mTOR (Raptor) (Cell signaling mAB 2280, USA, 1:1000), anti-p-Raptor ser722 (Merck-Millipore 09-104, Ireland, 1:1000), anti-p-Raptor ser792 (Cell Signaling 2083, USA, 1:1000), anti-phosphoglycerate kinase 1 (PGK) (Abcam ab199438, USA, 1:2000), and anti-Caspase 3 (Cell Signaling mAB 9665, USA, 1:1000).

**Compounds library and high throughput screening system**. A yeast-based, HTS system was established for the development of Fer-kinase inhibitors. This HTS was based on the previously described yeast growth inhibition assay by a mammalian kinase ectopically expressed in these cells[29]. To adopt this system for the identification of Fer inhibitors, we ectopically expressed Fer in the yeast strain BY4741-3702, which lacks the phosphatase encoding-*ptc1*gene. This led to the attenuation of the yeast growth rate. Inhibition of the Fer-kinase activity was expected to relieve the yeast growth attenuation. This yeast-based HTS system was implemented in a Tecan multi-functional robot, and applied for screening a library comprised of 100,000 synthetic compounds (ChemDiv, USA). The robotic system seeded the yeast cells in 96-well plates and administered each of the different tested compounds at ascending concentrations ranging between 0.1 and 50 µM. The seeded yeast cells were then allowed to grow for 40 h, and OD_{600}, was measured robotically by the ELISA reader installed in the system. Compounds which restored the growth of the Fer-expressing yeast cells by at least 30% were chosen, and their effective concentrations were determined in a second round of selection. The automated screening cycles followed by the selection of the effective compounds led to the identification of four hit compounds which demonstrated the highest restoration of growth profile of the Fer-expressing cells. These underwent an initial round of SAR analysis leading to the identification of two lead compounds, one of which was 0342.

A second round of SAR based on the structure of 0342 was conducted, thereafter leading to the development and synthesis of E260. The HTS system excludes Pan-interfering compounds, un-selective toxic compounds, and compounds that are unable to penetrate the membrane of intact live cells. After identification of Fer-inhibiting compounds, we further tested the cytotoxic effect of these compounds on normal somatic and cancer cells. To this end, the Tecan robotic system was adjusted to conduct 2,3-bis-(2-methoxy-4-nitro-5-sulfophenyl)-2h-tetrazolium⁻5-carboxanilide (XTT) based viability assay. This viability assay was based on the same method of cell inoculation in a 96-well plates and administration of the different Fer-inhibiting compounds in ascending concentrations ranging between 0.1 and 50 µM. The seeded cells were incubated with the tested compounds for 96 h. Following incubation, XTT dye (Life technologies, USA) was added according to the manufacturer's protocol. Following incubation to allow uptake of XTT, the relative fluorescence intensities were measured in each well using an ELISA reader installed in the system. After

complete data acquisition from the second HTS assay, we compared the results obtained from the two HTS systems and correlated the restoration of the yeast growth with the selective elimination of cancer cells. This cross analysis enabled us to successfully identify E260 as the best candidate for inhibiting the Fer-kinase activity and for selectively killing cancer cells. Notably, we found a direct correlation between the successful restoration of growth in Fer-expressing yeast cells and the effective and selective cytotoxic properties of the compounds in cancer cells. E260 was also computationally analyzed using the ChemSoft software (ChemDiv, USA) and proven to possess drug-likeness according to the Lipinski's rule of 5 [61].

**Chemical analyses of the newly developed compounds.** E260 was synthesized (Supplementary Fig. 1C) and used as a tartrate salt (compound **X** in Supplementary Fig. 1C) for the in vitro mammalian cell-cultures and in vivo experiments. The chemical identities of the synthetic intermediates were determined using $^{1}$H-NMR, HRMS, and LC–MS analyses (Supplementary Fig. 2 and below). The chemical identity and sample purity of the final synthesis product and the main SMC used in this study, E260-tartrate, (**X**) were determined using $^{1}$H-NMR, $^{13}$C-NMR, LC–MS, FTIR, and UPLC integrated with HRMS (Supplementary Fig. 1D–I and below).

NMR spectra were recorded, in Bruker instruments, for the following materials: **II** (in CDCl$_3$, $^{1}$H at 300 MHz, and $^{13}$C at 176.1 MHz at 300 K); **III** (in DMSO-$d_6$, $^{1}$H at 300 MHz, and in CD$_3$OD, $^{13}$C at 176.1 MHz at 300 K); **IV** (in DMSO-$d_6$, $^{1}$H at 300 MHz, and in CDCl$_3$, $^{13}$C at 176.1 MHz at 300 K); **Va** (in DMSO-D$_6$, $^{1}$H at 300 MHz, and in Acetone-$d_6$, $^{13}$C at 176.1 MHz. at 300 K); **VI** (in DMSO-D$_6$, $^{1}$H at 300 MHz, and in Acetone-$d_6$ $^{13}$C at 176.1 MHz at 300 K; **VIII** (in DMSO-$d_6$, $^{1}$H at 400.4 MHz, and in CDCl$_3$ $^{13}$C at 176.1 MHz at 300 K); **IX** (in CDCl$_3$, $^{1}$H at 400.4 MHz, and in CDCl$_3$ $^{13}$C at 176.1 MHz at 300 K); and **X** (in DMSO-$d_6$, $^{1}$H and $^{13}$C at 700.5 and 176.1 MHz, respectively, at 318 K). For **X**, a series of 2D spectra were also obtained (COSY-$^{1}$H × $^{1}$H correlation; HMQC-one-bond $^{13}$C × $^{1}$H correlation; and HMBC-long-range $^{13}$C × $^{1}$H correlation), which allowed the full attribution of all carbon and proton signals and confirms the chemical structure as shown.

LC–MS (HR) information: UPLC (Agilent 1260) integrated with HRMS were obtained on a 6545 QTOF instrument (Agilent) using the following parameters:

HPLC-Agilent 1260: Column: Agilent Zorbax HD-C18, 1.8-Micron, 2.1 × 50 mm, mobile phase A: water/0.1% formic acid, mobile phase B: acetonitrile/0.1% formic acid, gradient: 15–95% B in 10 min, hold 95% B to 2 min, 95% B to 5% in 2 min, 5% B for 2 min, then stop. Flow rate: 0.5 ml/min. Column temperature: 30 °C. Sample preparation: dissolved in water/acetonitrile (1:1). Detector: 210, 254, and 355 nm.

HRMS parameters: Interface:. ESI (positive), SheathGasFlow 12, SheathGasTemp (°C) 400, Nebulizer (psig) 40, Gas Flow (l/min) 8, Gas Temp (°C) 300, OctopoleRFPeak 750, Skimmer1 45, Fragmentor 180, Nozzle Voltage (V) 0, VCap 3500.

LC–MS information: HPLC-Shimadzu Analytical 10Avp series, autosampler: Gilson 215, ELSD: Sedex 75(55), mass spectrometer: PE SCIEX API 165(100). Column: Synergi 2 µ Hydro-RP Mercury, 20 × 2.0 mm. Mobile phase A: water/0.05% TFA. Mobile phase B: acetonitrile/0.05% TFA. Gradient: 0.01 min—5% B, 3.6 min—95% B, 4.5 min—95% B, 5.00 min—controller stop. Flow rate: 0.7 ml/min. Pressure range: 0–4500 psi. Sample preparation: dissolved in water/acetonitrile (1:1). Detector: 210, 254 nm. MS parameters: Interface: ESI (positive).

Fourier transform infrared spectroscopy: FTIR spectrum was collected using a Thermo Scientific Nicolet iS10 FTIR spectrometer equipped with a Smart iTR attenuated total reflectance sampler containing a single bounce diamond crystal. Data were collected and analyzed using OMNIC software. Spectra were collected in the 650–4000 per cm range at a spectral resolution of 4.

Melting point determination: The melting point (MP) of Compound **III** was determined using the Buchi melting point apparatus M-560 (BUCHI Corporation, UK) set to default parameters. A quick capillary-thermometer-hotplate based method was used for all other compounds. Briefly, small amount of finely powdered crystals of the test compound was placed in a thin-walled capillary tube. The tube was placed in very close proximity to a thermometer and capillary tube was heated in a small electric hotplate (IKA, Germany) in a rate of heating no more than 1° or 2° rise per minute. The capillary was closely observed until the crystals were completely melted to determine the MP. Differences in measured versus literature reported MP values could result from differences in determination methods and compounds purity.

**Compound II (Supplementary Fig. 1C): Tert-butyl 4-[(4-methylpiperazin-1-yl) carbonyl]piperidine-1-carboxylate.** $^{1}$H-NMR (CDCl$_3$) $\delta$: 4.23–4.05 (m, 2H), 3.65 (m, 2H), 3.54 (m, 2H), 2.77 (m, 2H), 2.62 (m, 1H), 2.41 (m, 4H), 2.33 (s, 3H), 1.80–1.60 (m, 4H), 1.46 (s, 9H); $^{13}$C-NMR (CDCl$_3$) $\delta$: 172.81 (C$_{5'}$), 154.64 (C=O BOC), 79.48 (C-BOC), 55.38 (C$_{8''a}$), 54.74 (C$_{8''b}$), 45.99 (C$_{10''}$), 45.33 (C$_{7''a}$), 43.56 (C$_{2''a}$), 42.83(C$_{2''b}$), 41.61(C$_{7''b}$), 38.32 (C$_{4''}$), 28.42 (Me-BOC), 28.39 (C$_{3''}$); HRMS (m/z): [MNa]$^{+}$ calcd. for C$_{16}$H$_{29}$N$_3$O$_3$, 334.21011; found, 334.20986. MP: 98 °C.

**Compound III (Supplementary Fig. 1C): 1-Methyl-4-(piperidin-4-ylcarbonyl) piperazine bis hydrochloride.** $^{1}$H-NMR (DMSO-$d_6$) $\delta$: 11.45 (s, 1H), 9.15 (s, 1H),

8.82 (s, 1H), 4.61–4.03 (m, 2H), 3.65–3.25 (m, 5H), 3.20–2.82 (m, 6H), 2.74 (s, 3H), 1.84–1.65 (m, 4H); $^{13}$C-NMR (CD$_3$OD) $\delta$: 174.13 (C$_{5'}$), 54.51 (C$_{8''a}$), 54.20 (C$_{8''b}$), 44.27 (C$_{2''}$), 43.69 (C$_{10''}$), 43.65 (C$_{7''a}$), 40.06 (C$_{7''b}$), 36.30(C$_{4''}$), 26.65 (C$_{3''b}$), 26.43 (C$_{3''a}$); HRMS (m/z): [M + H]$^{+}$ calcd. for C$_{11}$H$_{21}$N$_3$O, 212.17574; found, 212.1762. MP: 298 °C.

**Compound IV (Supplementary Fig. 1C): 1-Methyl-4-(piperidin-4-ylmethyl) piperazine.** $^{1}$H-NMR (DMSO-$d_6$) $\delta$: 3.30–2.92 (br, 2H), 2.90 (m, 2H), 2.39 (t, J=12Hz, 2H), 2.35–2.17 (m, 6H), 2.15 (s, 3H), 2.05 (d, 2H), 1.57 (m, 2H), 1.51 (m, 1H), 0.92 (m, 2H); $^{13}$C-NMR (CDCl$_3$) $\delta$: 65.28 (C$_{5'}$), 55.19 (C$_{8''a}$), 53.64 (C$_{7''}$), 46.64 (C$_{2''}$), 46.08 (C$_{10''}$), 33.80 (C$_{4''}$), 32.30 (C$_{3''}$); HRMS (m/z): [MH]$^{+}$ calcd. for C$_{11}$H$_{23}$N$_3$, 198.19647; found, 198.19623.

**Compound Va (Supplementary Fig. 1C): 2-Amino-1,3,4-thiadiazol.** $^{1}$H-NMR (DMSO-$d_6$) $\delta$: 8.55 (s, 1H), 7.18 (s, 2H);); $^{13}$C-NMR (Acetone-$d_6$) $\delta$: 168.98 (C$_5$), 143.16 (C$_2$); HRMS (m/z): [M + H]$^{+}$ calcd. for C$_2$H$_3$N$_3$S, 102.01204; found, 102.01207. MP: 185–187 °C (ref.[62]; 191 and 192 °C).

**Compound VI (Supplementary Fig. 1C): 2-Amino-5-bromo-1,3,4-thiadiazole.** $^{1}$H-NMR (DMSO-$d_6$) $\delta$: 7.51 (s, 2H); $^{13}$C-NMR (acetone-$d_6$) $\delta$: 171.57 (C$_5$), 125.23 (C$_2$); HRMS (m/z): [M + H]$^{+}$ calcd. for C$_{15}$H$_{18}$BrNO, 179.92256 and 181.92042; found, 179.92289 and 181.92066 (1:1 doublet). MP: 192–193 °C (ref [63];180 and 181 °C).

**Compound VIII (Supplementary Fig. 1C): 2-Bromo-6-(4-isopropylphenyl) imidazo[2,1-b][1,3,4]thiadiazole.** $^{1}$H-NMR (DMSO-$d6$) $\delta$: 8.68 (s, 1H, H$_5$), 7.77 (d, J=8 Hz, 2H, H$_{2'}$), 7.28 (d, J=8 Hz, 2H, H$_{3'}$), 2.90 (m, 1H, H$_{5'}$), 1.22 (d, J=7 Hz, 6H, H$_{6'}$); $^{13}$C-NMR (CDCl$_3$) $\delta$: 149.32 (C$_{4'}$), 145.39 (C$_6$), 145.31 (C$_8$), 133.82 (C$_2$), 129.76 (C$_{1'}$), 127.03 (C$_{3'}$), 125.28 (C$_{2'}$), 109.54 (C$_5$), 33.95 (C$_{5'}$), 23.90 (C$_{6'}$) MS (ESI$^+$) (m/z) [MH]$^+$:calcd. for C$_{13}$H$_{12}$BrN$_3$S 322.2, 323.9; found 322.2, 324.0; HRMS (m/z): [MH]$^{+}$ calcd. for C$_{13}$H$_{12}$BrN$_3$S, 322.00081; found, 322.00047. MP: 139–141 °C.

**Compound IX (Supplementary Fig. 1C): 6-(4-isopropylphenyl)-2-{4-[(4-methylpiperazin-1-yl)methyl]piperidin-1-yl]imidazo[2,1-b][1,3,4]thiadia-zole.** $^{1}$H-NMR (CDCl$_3$) $\delta$: 7.71 (m, 3H, H$_5$, H$_{2'}$), 7.26 (d, J=8 Hz, 2H, H$_{3'}$), 3.87 (d, J=12 Hz, 2H, H$_{2''}$), 3.09 (t, J=12 Hz, 2H, H$_{2''}$), 2.93 (sept, J=6.5 Hz, 1H, H$_{5'}$), 2.50 (br, 8H, H$_{7''}$, H$_{8''}$), 2.34 (s, 3H, H$_{10''}$), 2.25 (d, J=6.8 Hz, 2H, H$_{5''}$), 1.91 (d, J=12.2 Hz, 2H, H$_{3'}$), 1.79 (m, 1H, H$_{4'}$), 1.33 (d, J=12.2 Hz, 2H, H$_{3'}$), 1.29 (m, 6H, H$_{6'}$); $^{13}$C-NMR (CDCl$_3$) $\delta$: 164.59 (C$_2$), 147.50 (C$_{4'}$), 143.56 (C$_6$), 141.30 (C$_8$), 132.09 (C$_{1'}$), 126.64 (C$_{3'}$), 124.57 (C$_{2'}$), 108.58 (C$_5$), 64.05 (C$_{5''}$), 55.15 (C$_{8''}$), 53.63 (C$_{7''}$), 48.45 (C$_{2''}$), 46.05 (C$_{10''}$), 33.83 (C$_{5'}$), 33.15 (C$_{4'}$), 29.76 (C$_{3'}$), 23.98 (C$_{6'}$); MS (ES$^+$) (m/z): [MH]$^{+}$ calcd. for C$_{24}$H$_{34}$N$_6$S 439.0; found, 439.5;); HRMS (m/z): [MH]$^{+}$ calcd. for C$_{24}$H$_{34}$N$_6$S 439.26384; found, 439.26346. MP: 190–192 °C.

**Compound X (Supplementary Fig. 1C): 6-(4-isopropylphenyl)-2-{4-[(4-methylpiperazin-1-yl)methyl]piperidin-1-yl]imidazo[2,1-b][1,3,4]thiadiazole tartrate (salt).** $^{1}$H-NMR (DMSO-$d_6$) $\delta$: 8.17 (s, 1H, H$_5$), 7.68 (d, J=8 Hz, 2H, H$_{2'}$), 7.22 (d, J=8 Hz, 2H, H$_{3'}$), 4.14 (s, 2H, CHOH-tartrate), 3.87 (d, J=13 Hz, 2H, H$_{2''}$), 3.14 (t, J=13 Hz, 2H, H$_{2''}$), 2.88 (sept, J=7 Hz, 1H, H$_{5'}$), 2.69 (m, 4H, H$_{8''}$) 2.49 (m, 4H, H$_{7''}$), 2.42 (s, 3H, H$_{10''}$), 2.22 (d, J=7 Hz, 2H, H$_{5''}$), 1.81 (d, J=12 Hz, 3H, H$_{3'}$, H$_{4'}$), 1.29 (m, 2H, H$_{3'}$), 1.21 (d, J=7 Hz, 6H, H$_{6'}$) $^{13}$C-NMR (DMSO-d6) $\delta$: 173.36 (COOH), 164.14 (C2), 146.68 (C$_{4'}$), 142.48 (C6), 140.40 (C8), 132.01 (lC1'), 126.29 (C3'), 124.14 (C2'), 109.35 (C5), 71.69 (CHOH-tartrate), 62.72 (C$_{5''}$), 53.75 (C8''), 51.58 (C7''), 44.13 (C10''), 48.22 (C2''), 33.03 (C$_{5'}$), 32.07 (C4''), 29.05 (C3''), 23.74 (C$_{6'}$); HRMS (m/z): [MH]$^{+}$ calcd. for C$_{24}$H$_{34}$N$_6$S,439.263; found, 439.263. MP: 206–207 °C.

**Synthesis process of E260 as a tartrate salt.** The following compounds were synthesized according to the scheme in Supplementary Fig. 1C.

**Compound II: Tert-butyl 4-[(4-methylpiperazin-1-yl)carbonyl]piperidine-1-carboxylate.** 218 mmol (50.00 g) of compound **I** were stirred with 218 mmol (35.36 g) of CDI in 500 ml chloroform for 1 h at room temperature. Then, 218 mmol (21.84 g) 1-methylpiperazine were added and the reaction mixture was stirred under reflux for 2 h. When the reaction mass was cooled down to RT, 200 ml water were added, phases were separated, organic phase was washed with water (2 × 100 ml), combined organic layers were dried under anhydrous Na2SO4, and the solvent was removed under reduced pressure to provide 66.55 g (98%) of compound **II**, as a colorless oil.

**Compound III: 1-Methyl-4-(piperidin-4-ylcarbonyl)piperazine bis hydrochloride.** 217 mmol (67.57 g) of **II** were dissolved in 400 ml EtOH and 36 ml of concentrated hydrochloric acid were added. The reaction mass was then stirred under reflux for 2 h. When cooled down, the formed precipitate was removed by

filtration, washed with EtOH and air dried to provide 36.40 g (59%) of compound **III** which was obtained as a white solid substance.

**Compound IV: 1-Methyl-4-(piperidin-4-ylmethyl)piperazine**. 412.3 mmol (15.649 g) LiAlH4 were suspended in 450 ml of absolute THF, then 126.9 mmol (36.06 g) of compound **III** were slowly added and the reaction mixture was refluxed for 48 h. When the reaction mixture was cooled down, 16 ml water were added drop wise, followed by the addition of 16 ml 15% water solution of NaOH and final addition of 48 ml water. The reaction mixture was then filtered and the solid product was washed twice with 250 ml THF. Solvent was removed under reduced pressure to provide 17.77 g (71%) of compound **IV**, which was obtained as a colorless oil.

**Compound Va: 2-Amino-1,3,4-thiadiazol**. 91.5 g (1 mol) of thiosemicarbazide (**V**) were suspended in 50 ml of 85% formic acid in water solution, followed by the careful addition of 80 ml sulfuric acid. The reaction mixture was stirred for 3 h on a boiling water bath and was then left to cool down. 800 ml water were then added and the precipitate was filtered off, washed with water, and dried. A yield of 100 g (99%) 2-amino-1, 3, 4-thiadiazole (**Va**), was obtained.

**Compound VI: 2-Amino-5-bromo-1,3,4-thiadiazole**. A suspension of 40.0 g (0.396 mole) 2-amino-1, 3, 4 –thiadiazole (**Va**) and 156.07 g (1.902 moles) sodium acetate was prepared in 460 ml of glacial acetic acid. The mixture was stirred and 69.53 g. (0.435 mole) bromine, dissolved in 100 ml. glacial acetic acid, were added drop wise. The reaction mixture developed an orange color during the addition of the bromine.

The reaction mixture was then heated for 1 h on a steam bath. At the end of this step, the reaction mixture was poured onto an ice cooled 2 L volume glass and the precipitated solid substance was recovered by filtration. The solid product was re-suspended in water and the mixture was adjusted to pH 11 with aqueous ammonium hydroxide. The obtained product was filtered off on a Buchner funnel, washed with water, and dried to yield 35 g (49%) 2-amino-5-bromo-1,3,4-thiadiazole (compound **VI**).

**Compound VIII: 2-Bromo-6-(4-isopropylphenyl)imidazo[2,1-*b*][1,3,4]thiadiazole**. 59.93 mmol (10.79 g) 2-amino-5-bromo-1,3,4-thiadiazole (**VI**) were suspended in 180 ml absolute EtOH, then 70.5 mmol (17.0 g) 2-bromo-4′-isopropylacetophenone (**VII**)) were added and the reaction mixture was stirred overnight at RT. Then the mixture was stirred for 12 h at 60 °C, followed by additional stirring for 24 h at 78 °C under reflux. A fraction ( ~ 80 ml) of the solvent was removed under reduced pressure, to achieve a doubled concentration of the mixture. The product was filtered off, washed with 15 ml of cold EtOH, and dried to provide a pure compound **VIII** (12 g, 52.8 %), which was obtained as a white solid substance.

**Compound IX: 6-(4-isopropylphenyl)-2-{4-[(4-methylpiperazine-1-yl) methyl]piperidin-1-yl}imidazo[2,1-*b*][1,3,4]thiadiazole**. 13.68 mmol (4.41 g) of compound **VIII** were suspended in 150 ml absolute isopropanol. Then 30.41 mmol (3.93 g) diisopropylethylamine (DIPEA), and 15.2 mmol (3.0 g) 1-methyl-4-(piperidin-4-ylmethyl) piperazine (**IV**) were added and the reaction mixture was heated at 85° C under reflux, overnight. After cooling down, the product was filtered off, washed with 5 ml EtOH, then washed twice with 15 ml acetone, and dried. This yielded 4.2 g (63 %) of compound IX which was obtained as a white solid substance.

**Compound X: 6-(4-isopropylphenyl)-2-{4-[(4-methylpiperazine-1-yl)methyl] piperidin-1-yl}imidazo[2,1-*b*][1,3,4]thiadiazole, (2*R*,3*R*)-2,3-dihydroxybutanedioate (1:1-tartrate salt)**. To a hot (50 °C) solution containing 7.3 mmol (3.2 g) **IX** in 150 ml EtOH, a hot (50 °C) solution comprised of 7.3 mmol (1.1 g) tartaric acid in 30 ml EtOH, was added with stirring. The reaction was then heated for 30 min at 78 °C under reflux. After cooling to RT the white precipitate was filtered off, washed with EtOH and dried. This yielded 4 g, ( > 95 %) of the product **X** which was obtained as white solid substance.

**In vitro kinase inhibition assay**. The Fer/FerT KD protein fragment was expressed in bacterial cells as described above. It was then purified twice using the Talone cobalt affinity resin beads (Clontech, USA) and eluted using a 10-400 mM imidazole gradient. The protein was further purified using a 5000 Dalton cut-off dialysis and concentrated using a 10,000 Dalton cut-off Amicon Ultra-4 centrifugal filter (Merck-Milipore, Ireland). Following purification, a sample of the purified protein was separated onto SDS–PAGE and subjected to a WB analysis using specific antibodies directed towards the Fer/FerT KD, to verify the integrity of the purified protein. To test the effect of E260 on the Fer/FerT KD autophosphorylation activity, 0.5 µg of the Fer/FerT KD protein was incubated in 0.5 ml kinase activity buffer (50 mM HEPES pH 7.5, 10 mM MgCl₂, 1 mM EGTA, 0.01% Brij-35) and 1 µM ATP (Fermentas, USA). As a negative control, the KD protein was incubated in the same buffer without ATP. The KD and ATP containing mixture was incubated for 1 h at room temperature with ascending concentrations

of E260 dissolved in DMSO or with DMSO alone. Following the incubation period, a sample from the incubated mixture was separated by SDS–PAGE and a WB analysis was performed using specific anti-Fer and anti-pY antibodies to evaluate the inhibitory effect of E260, as reflected by the diminished phosphorylation level of the Fer/FerT KD.

**Computational modeling of the docking of E260 in Fer/FerT**. Homology modeling of a Fer fragment extending from residues 447-820, including the enzymes SH2 and KD, was performed using the MODELER protocol in Discovery Studio version 4.0 (Accelrys Inc, San Diego, USA). As a template, we used the structure of the corresponding domains in the Tyrosine-protein kinase Fes/Fps, PDB 3BKB chain A, with 66% identity, 80% similarity, and 1% gaps. Docking of E260 to Fer/FerT was performed using AutoDock Vina[64] with default parameters. The binding site was defined as the whole-protein structure, and 20 binding modes were suggested (scores ranging from 6 to 7.8). For our analysis, we used the highest scoring pose (score = 7.8). Notably the second best pose (score = 7.3) was also positioned in the same cavity.

**Analysis of E260 and the Fer KD interaction**. The Fer recombinant KD was produced and purified as described above and was labeled with a reactive dye NT-647 using N-hydroxy succinimideester chemistry using the manufacturer kit (NanoTemper Technologies, Munich, Germany). E260 was dissolved in DMSO and incubated with NT-647-labeled Fer KD or DMSO in binding buffer (20 mM Tris-HCl, pH 7.5, 150 mM NaCl, and 0.01% Nonidet P-40) at concentrations ranging between 25 nM and 25 µM. The binding of E260 to the labeled Fer KD was measured by using the microscale thermophoresis technology in a Monolith NT.115 reader (NanoTemper) as detailed previously[65] using the thermophoresis and Temperature Jump analysis. The dissociation constant ($K_d$) of the binding was calculated using nonlinear regression analysis program embedded in the device software.

**Dynamic light scattering (DLS) measurements of micelles**. Emulsification of E260 was performed in Cremophor EL and PBS mixture. Photon Cross-correlation Spectroscopy measurements combined with Pade–Laplace analysis was carried out and the size distribution and stability of the micelles were determined. All analyses were carried out using NanoPhox (Sympatec, Germany).

**Immunoprecipitation**. Whole-cell protein lysates (1 mg, for the Fer and FerT immunoprecipitations) or nuclear proteins (5 mg, for Fer-PARP-1 co-immunoprecipitation experiment) were prepared from SW620 CC cells. These were taken for immunoprecipitation of Fer or FerT using specific antibodies as follows: Protein lysates (5 mg) were incubated overnight at 4 °C with 1:100 diluted antibodies. Antigen–antibody complexes were precipitated with protein A Sepharose (GE Healthcare 17-0780-01) after 1.5 h incubation at 4 °C. Precipitates were washed twice with Tris buffer, pH 7.5, containing 20 mM Tris-HCl, 10% glycerol, EDTA 1 mM, 0.1% Triton X-100, 150 mM NaCl, and once with 75 mM NaCl. Recovered immunocomplexes were solubilized in Laemmli sample buffer and were separated on SDS–PAGE. The protein was visualized by western blotting using specific antibody.

**Analyzing the inhibition of Fer by E260 in cells**. Deliberate activation of Fer which leads to its auto-phosphorylation was carried out as described by applying 16 h serum starvation followed by 20 min incubation of cells with 3 mM H2O2 [31].

Fer was immunoprecipitated from cells lysates (1 mg) for 2 h at room temperature using protein A Sepharose beads which were conjugated to antibodies directed toward Fer's N′ terminus. The eluted precipitates were resolved on SDS–PAGE, and proteins were visualized by WB using specific antibodies.

**Cell survival assay**. Cells death level was determined using the MultiTox-Fluor Multiplex Cytotoxicity Assay (G9201, Promega, USA) according to the manufacturer's protocol. Briefly, cells were inoculated into black 96-well plate. After 24 h, when the cells were completely attached, E260 or control solution were administrated at different concentrations and incubated for the desired period of time. Following the incubation period, the assay's fluorophore which was used to determine the cell death levels was added to each well. The relative fluorescence intensity emitted by the fluorophore from each well was determined using an ELISA reader and was compared to the florescence intensity obtained from the standard curve drawn according to the manufacturer's protocol to translate it to cell death percentage and normalized to the non-treated cells which were also used in each analysis.

**HMGB1 release assay**. Concentration of the HMGB1 protein in the cells growing medium was measured according to the manufacturer's instructions using the HMGB1 ELISA kit (OKEH00423, Aviva systems biology, USA).

**Fluorescence-activated cell sorter viability assay**. Cell death was determined using the Annexin V and PI assay (Biovision,USA) as described[7].

**Transmission electron microscopy**. Cells were grown in MEM or DMEM, washed twice with PBS and fixed for 5 min in Karnovsky fixative (2.5% glutar-aldehyde, 2.5% paraformaldehyde in cacodylate buffer pH 7.4) on the tissue culture plate. Cells were then collected, and transferred to 1.5 ml tube for further 1 h fixation at room temperature. The cells were then stored O.N at 4 °C after which the samples were washed in 0.1 M cacodylate buffer and post-fixed with 1% OsO4 in 0.1 M sodium cacodylate buffer (Sigma-Aldrich, Israel) for 1 h. Samples were then dehydrated in alcohol and propylene oxide followed by embedding in Agar Mix. Thin sections (60 nm) were cut, stained with uranyl acetate and lead citrate, and observed in a FEI Tecnai transmission electron microscope.

**Determination of the mitochondrial membrane potential**. For microscopic and ELISA-based analyses of the MMP, tetra-methyl-rhodamine-ethyl ester incor-poration assay was carried out according to the manufacturer's instructions (Abcam, USA).

**High pressure liquid chromatography quantification**. Quantification of intra-cellular ATP was carried out using high pressure liquid chromatography (HPLC) as previously described[7].

**Measurement of the mitochondrial OXPHOS complex I activity**. Activity of the mitochondrial OXPHOS complex I (NADH dehydrogenase) was determined using the Microplate Assay for Human Complex I Activity (MS141-MitoSciences), according to the manufacturer's instructions and as described[7].

**Immunocytochemical analyses**. Immunocytochemical analyses of cells treated with E260 or control solution were carried out as previously described[7].

**Interactome analysis**. Interactome analysis of Fer kinase was carried out as pre-viously described[9]. Briefly, Fer was immunoprecipitated from SW620 CC cells, followed by protein separation on SDS–PAGE and LC–MS/MS analyses.

**siRNA-mediated silencing of Fer**. siRNA-mediated silencing of Fer in SW620 cells was carried out using Lipofectamine2000 as previously described (Invitrogen, USA)[7]. The applied siRNA sequences were:

1. siRNA-fer 5′-ACGUAUCCAAGUCUUGGCUACUUAU-3′ 2. siRNA-fer 5′-GGAAUUACGGUUACUGGAAACAGUA-3′ For negative control (siRNA-neg)–Stealth RNAi siRNA Negative Control Med GC (Invitrogen, USA) was used.

**Terminal deoxynucleotidyl transferase assay**. Samples containing $1 \times 10^6$ cells were inoculated into a 6-well plate containing a cover slip, and fixed for 30 min with 4% paraformaldehyde following adhesion. The cells were then subjected to TUNEL analysis according to the manufacturer's instructions (Roche, USA).

**Pharmaco-kinetic study of E260**. All animal experiments were performed according to the guidelines of the Institutional Animal Care and Use Committee of the Bar-Ilan University. The investigation of the pharmaco-kinetic properties of E260 was carried out using 30 ICR/CD-1 mice. Intraperitoneal (IP) administration was used, and the solution was administered as 10 ml per 1 kg of animal body weight.

At 5, 15, 30, 60, 120, and 240 min after administration (five mice per each time point) mice were decapitated and blood samples were taken in EDTA-containing tubes. Plasma was prepared from the collected blood by centrifugation and then forwarded for LC–MS/MS analysis. A high sensitivity analytical method was developed for detection of E260. Analysis was performed in the Agilent 1100 HPLC system including degasser, the binary pump, autosampler, and absorption detector with variable wavelength ranging between 190 and 600 nm. The HPLC system was coupled with the tandem mass spectrometer API 2000 (Applied Biosystems). The TurboIonSpray ion source was used in the positive ion mode. Data acquisition was performed with Analyst 1.3.1 software.

**Quantification of glycolytic rate of cells**. Cells were grown in MEM containing 1 g/l glucose or in DMEM with 4.5 g/l glucose. Secreted lactate, which correlates with the glycolytic rate, was determined using Lactate Colorimetric Assay Kit II according to the manufacturer's protocol (Biovision, USA).

**Animals xenograft studies**. All animal experiments were performed according to the guidelines of the Institutional Animal Care and Use Committee of the Bar-Ilan University. Nude mice (Harlan, Israel) were inoculated with $1.5 \times 10^6$ SW620 or SW48 CC cells and divided randomly into experimental groups as previously described[7]. The mice were transferred from ad libitum diet to a stricter diet 2 days before inoculation to lower blood glucose levels. At this point the mice were also housed one per cage to ensure an even consumption of food. The diet was

comprised of 3 g/mouse/day of standard chow, given at the same time every day. The food was consumed within an average time of 2 h, consequently the mice were kept without food for the next 22 h until the daily ration. The mice were kept on this diet throughout the experiment. The mice were randomized 4 days after tumor inoculation and placed again each in a cage. Mice were injected intraperitoneally every 12 h for 22 days with 25 or 50 mg/kg of the micellar E260 formulation, and control mice were injected with empty micelles.

**Toxicological and histopathological studies**. Mice were sacrificed at day 22 of the xenograft experiments, as described above. Tissues from relevant organs and tumors were fixed in paraffin, sectioned (7 µm sections) and stained with hematoxylin and eosin stain for histopathological evaluation as previously described[22]. Blood samples were drawn by cardiac puncture. The blood was allowed to clot for 30 min at room temperature and serum separation was carried out by 20 min centrifugation at $1000 \times g$. The serum was analyzed immediately using the Cobas 6000-501 Chemistry Analyzer (Roche Diagnostics, USA) according to the manufacturer's protocols for each assay.

**Ethidium homo-dimer III incorporation assay**. Onset of necrosis was monitored using the Ethidium homo-dimer III (EthD III) incorporation assay, according to the manufacturer's protocol (Biotium, USA). EthD III staining in necrotic cells or Hoechst staining in live intact cell were visualized immediately by Axioimager z1 fluorescent microscope (Zeiss, German).

**Statistical analysis**. Statistical analysis was performed using the paired and unpaired Student's t-tests, with a $P < 0.05$ being considered significant. Results are depicted as mean ± standard deviation (±SD) or ±standard error (±SE) of the mean for n given samples. For statistical analysis of groups with multiple comparisons, one-way ANOVA analysis was performed with Bonferroni post hoc, with $P < 0.05$ considered significant.

**Data availability**. The authors declare that all the other data supporting the findings of this study are available within the article and its Supplementary Information Files and from the corresponding author upon reasonable request.

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

## Acknowledgements

This work was supported by grants from the Calb, Luis Sheinman, and Milstein Foundations. We would like to thank Dr Michal Afri, Dr Michal Weitman, Dr Michal Ejgenberg, and Dr Hugo E. Gottlieb from the Chemistry Department at Bar-Ilan

university for their kind help with the chemical characterizations of the new entity SMCs described in this article.

## Author contributions

Y.E., E.Y., M.C., S.S., T.F., and E.H. performed the biological and chemical experiments. Y.E., S.S., and U.N. designed the experiments, analyzed the data and wrote the article. A.N. and Y.E. performed the toxicological evaluations in the treated animals. Y.E., A.F. and Y.O. performed the computational analyses.

## Additional information

**Competing interests:** Y.E., E.Y., M.C., S.S., and U.N., hold several patents on the Fer/FerT kinase inhibitors and their use. The remaining authors declare no competing financial interests.

