## [Peer Review File · Nature Communications]

Reviewers' comments:

Reviewer #1 (Remarks to Author):

This manuscript reports the discovery of a new compound E260 which binds to Fer and FerT and disrupts the binding of these tyrosine kinases to PARP-1. The details of the effects of E260 on the energetics of normal and tumour cells is investigated and antitumour activity is demonstrated in a model *in vivo*, without apparent toxicity. The manuscript is generally well written, although some parts are more difficult for the non-specialist to follow. The conclusions are justified by the data and the work is of sufficient impact for publication in *Nature Communications*.

There are one major point and several minor points that the authors should address in revising their manuscript towards to final acceptable version:

1. *Major point.* This manuscript is the first report of a new bio-active small molecule. Currently, there are no descriptions or references as to how this compound was synthesised and its identity and purity were proven. A literature search revealed that the first and only disclosure of the compound was in a patent by some of the authors [Nir, U.; Shpungin, S.; Yaffe, E.; Cohen, M. WO 2010/97798 A1] and this patent does not give full details. It is essential that the synthesis of the key compound in the work should be fully described and the spectroscopic characterisation provided in the journal literature, to allow the reader to judge the quality of the reagent. These could be provided in the Supporting Information.
2. *Minor point.* There are minor inconsistencies in referencing styles.
3. *Minor point.* Attention should be paid to proper punctuation when using adjectival and adverbial nouns.
4. *Minor point.* In the Introduction, the authors note that Fer is present in normal cells and the truncated variant FerT is present in sperm cells and in cancer cells. What is the position in normal stem cells, especially the haematopoietic cells in the bone marrow? This should be clarified. Moreover, in the anti-cancer studies *in vivo*, no loss of weight was observed with E260 but was there any effect on the bone marrow? This is a likely site of toxicity.
5. *Minor point.* On page 11, the authors use 3-aminobenzamide as a “PARP-1 inhibitor”. Whereas this was the gold standard for inhibition of PARP-1 around 25 years ago, it is now known as a relatively weak inhibitor (IC_{50} ca. 20 μ M) and to be non-selective between the multiple PARP isoforms. It is distinctly possible that collateral inhibition of other PARPs is occurring, contributing in part to the observed results.
6. *Minor point.* The correct abbreviation for gram is g, not gr.

Reviewer #2 (Remarks to the Author):

In the presented manuscript, Elkis et al. identify the novel Fer-Kinase-Inhibitor E260 and provide evidence about its therapeutic potential in vivo as it "selectively" induces cell death in tumor cell lines, but spares healthy cells. Although this is a well-conducted study and generally should be considered for publication in Nature Communications, some concerns should be addressed to further improve this manuscript. Especially, there have been a number of recently defined pathways of regulated necrosis that have not been discussed or investigated.

Major concerns:

- The authors completely fail to identify this pathway of cell death in the light of the current literature (e.g. Vanden Berghe and Linkermann et al., Nature Reviews MCB 2014). Identification of PARP1 (Fig.5) as key player of necrosis (SuppFig. 3) with E260 is not sufficient to conclude anything about parthanatos. It is missing which pathways of regulated necrosis are actually active. Additional experiments with small molecule inhibitors, such as Nec-1s (Necroptosis) or Fer-1 (Ferroptosis) and even zVAD (Apoptosis) combined with E260 should be performed to clarify this issue. The breakdown of mitochondrial membrane potential is not hallmark only of Parthanatos/MPT-RN, but also the other ones later during cell death which cannot be differentiated by the provided assays.
- TEM pictures of necrotic cell death in Fig. 3 are fine, but TEM is always "artificial". The authors should provide a time-lapse video of the cell undergoing regulated necrosis as induced by E260. This actually would strengthen evidence and might give us a better understanding of the particular events during this program.

Minor remarks:

- Throughout the paper, the authors must indicate what they measured, not what they interpreted this to be like, in the plots (e.g. Fig. 2O). "Cell death" is an interpretation. TUNEL-positivity or PI-positivity in percent is what can be measured.
- Induction of autophagy in Fig. 4 is interesting, but this referee is not sure if inhibition of autophagy really protects - as Fig. 4K (12h) and 4L (16h, why using different times?) - in a longer time scale. Possibly, this protection simply displays a delay as ATP is not consumed quickly, but might be detrimental in the longer run. Time points for 24h/48h would be of interest.
- Fig. 3 O-T, Fig. 4 A-B, Fig. 5 D-E and Fig. 6 C-D all utilize the difference between compare SW620 (metastatic colon cancer) to Hfb cells. It might be possible that observed differences might result from different tissue origins rather than healthy and neoplastic. This possible bias should be addressed by providing another cell line - not for all experiments, but for some of the important ones.
- I wonder how E260 targets only cancer cells. This should be discussed in more detail, beyond metabolism. I'd be very interested in an experiment such as an ischemic model: Does E260 induce more cell death in these "diseased" cells?

Reviewers' comments:

Comments of Reviewer I (Cited point by point with the authors' response attached to each comment).

This manuscript reports the discovery of a new compound E260 which binds to Fer and FerT and disrupts the binding of these tyrosine kinases to PARP-1. The details of the effects of E260 on the energetics of normal and tumour cells is investigated and antitumour activity is demonstrated in a model in vivo, without apparent toxicity. The manuscript is generally well written, although some parts are more difficult for the non-specialist to follow. The conclusions are justified by the data and the work is of sufficient impact for publication in Nature Communications.

There are one major point and several minor points that the authors should address in revising their manuscript towards to final acceptable version:

- 1. Major point:** This manuscript is the first report of a new bio-active small molecule.

Currently, there are no descriptions or references as to how this compound was synthesised and its identity and purity were proven. A literature search revealed that the first and only disclosure of the compound was in a patent by some of the authors [Nir, U.; Shpungin, S.; Yaffe, E.; Cohen, M. WO 2010/97798 A1] and this patent does not give full details. It is essential that the synthesis of the key compound in the work should be fully described and the spectroscopic characterisation provided in the journal literature, to allow the reader to judge the quality of the reagent. These could be provided in the Supporting Information.

Authors' response: Following the reviewer's suggestion we include in the revised manuscript newly added **Figures- S1C and D** describing the synthesis scheme of the E260 compound and the ¹H-NMR spectra of the synthesized compound, respectively. This analysis confirmed the > 95 % purity of the E260 compound routinely used in our reported and presented experiments.

- 2. Minor point:** There are minor inconsistencies in referencing styles.

Authors' response: Following the reviewer's comment we have corrected all inconsistencies in the referencing styles and arranged the references according to the latest referencing style of *Nature Communications*, using the EndNote referencing program.

- 3. Minor point:** Attention should be paid to proper punctuation when using adjectival and adverbial nouns.

Authors' response: Following the reviewer's comment we sent the manuscript for re-editing by an experienced native-English speaking editor, with extra emphasis on grammar

and punctuation. The amendments were introduced into the revised manuscript and we now believe that it is more fluently read.

- 4. Minor point:** In the Introduction, the authors note that Fer is present in normal cells and the truncated variant FerT is present in sperm cells and in cancer cells. What is the position in normal stem cells, especially the haematopoietic cells in the bone marrow? This should be clarified. Moreover, in the anti-cancer studies in vivo, no loss of weight was observed with E260 but was there any effect on the bone marrow? This is a likely site of toxicity.

Authors' response: Following the reviewer's suggestion, we have checked the presence of FerT in normal human hematopoietic CD34⁺ stem-cells and examined the effect of E260 on these cells. As can be seen in the newly added **Figure S2C**, FerT is not expressed in these hematopoietic cells. The newly added **Figure S2B** shows that E260 exerts minor cytotoxic effect on these hematopoietic stem cells following 24h treatment, and this effect is slightly increased after 48h exposure to the compound. Although in comparison to normal human fibroblasts and epithelial cells it seems that E260 does exert some marginal cytotoxic effect on these cells, it should be noted that these primary cells are extremely difficult to propagate in culture and require therefore special growing medium (CellGro® SCGM medium, Cellgenix, Germany). Thus, some of the death evoked by E260 after 24h or 48h in culture may reflect the general vulnerability of the CD34⁺ cells when being transferred to the MEM medium to allow standardization of the comparative E260 experiments carried out with the different cell-types and cell lines.

- 5. Minor point:** On page 11, the authors use 3-aminobenzamide as a “PARP-1 inhibitor”.

Whereas this was the gold standard for inhibition of PARP-1 around 25 years ago, it is now known as a relatively weak inhibitor (IC₅₀ ca. 20 μM) and to be non-selective between the multiple PARP isoforms. It is distinctly possible that collateral inhibition of other PARPs is occurring, contributing in part to the observed results.

Authors' response: Following the reviewer's comment we clarify in the revised Discussion section of the manuscript that 3-aminobenzamide (3AB) is not a specific inhibitor of PARP-1 but can also inhibit other PARP isoforms like PARP-2.

- 6. Minor point:** The correct abbreviation for gram is g, not gr.

Authors' response: Following the reviewer's comment, we have corrected the abbreviation for gram to g.

Comments of reviewer II (Cited point by point with the authors' response attached to each comment).

:

Major concerns:

1. The authors completely fail to identify this pathway of cell death in the light of the current literature (e.g. Vanden Berghe and Linkermann et al., Nature Reviews MCB 2014). Identification of PARP1 (Fig.5) as key player of necrosis (SuppFig. 3) with E260 is not sufficient to conclude anything about parthanatos. It is missing which pathways of regulated necrosis are actually active. Additional experiments with small molecule inhibitors, such as Nec-1s (Necroptosis) or Fer-1 (Ferroptosis) and even zVAD (Apoptosis) combined with E260 should be performed to clarify this issue. The breakdown of mitochondrial membrane potential is not hallmark only of Parthanathos/ MPT-RN, but also the other ones later during cell death which cannot be differentiated by the provided assays.

Authors' response: Following the reviewer's comment we have extended our analysis to further characterize the type of death evoked by E260 in malignant cells. According to the reviewer's suggestion we have subjected Colon carcinoma (CC) cells to E260 in the presence of selective cell-death inhibitors and the effect of these inhibitors on the cytotoxic activity of E260 in malignant cells was determined. While Z-VAD (inhibitor of apoptosis) and Ferrostatin-1 (inhibitor of Ferroptosis) did not affect the cytotoxic effect exerted by E260, Nec-1(inhibitor of Necroptosis) did decrease the death level evoked by E260 by approximately 30%. Thus, the programmed form of necrosis-Necroptosis is involved in the selective cellular death evoked by E260 in malignant cells. These experimental results are presented in the newly added **Figures S3 K-M**, and are discussed in the revised Discussion section.

2. TEM pictures of necrotic cell death in Fig. 3 are fine, but TEM is always "artificial". The authors should provide a time-lapse video of the cell undergoing regulated necrosis as induced by E260. This actually would strengthen evidence and might give us a better understanding of the particular events during this program.

Authors' response: Following the reviewer's suggestion we have adopted an additional analytical approach for characterizing the onset of necrosis. Although not based on live microscopy as the reviewer suggested, we believe that this approach reliably enables us to

follow the kinetics of necrosis onset in E260 treated malignant cells and to correlate it with the TEM images depicting necrosis in **Figures 3A-H** of the revised manuscript. To this end we have consecutively followed for 24h the release of the necrosis marker- High mobility group box 1 (HMGB1) protein¹, from E260 treated CC cells. As can be seen in the newly added **Figures 3Y and Z**, the kinetic of HMGB1 release correlated well with the onset time of necrosis as is depicted by TEM images of E260 treated cancer cells (**Figures 3A-H**).

Minor remarks:

1. Throughout the paper, the authors must indicate what they measured, not what they interpreted this to be like, in the plots (e.g. Fig. 2O). "Cell death" is an interpretation. TUNEL-positivity or PI-positivity in percent is what can be measured.

Authors' response: Following the reviewer's comment we have now indicated in the revised Figures 2-6 and the supplementary Figures S2-S4, the actual cell-death or MMP reflecting parameters which were measured in each experiment.

2. Induction of autophagy in Fig. 4 is interesting, but this referee is not sure if inhibition of autophagy really protects - as Fig. 4K (12h) and 4L (16h, why using different times?) - in a longer time scale. Possibly, this protection simply displays a delay as ATP is not consumed quickly, but might be detrimental in the longer run. Time points for 24h/48h would be of interest.

Authors' response: In response to the reviewer's question we would like to clarify that the antagonistic effect of 3MA on the decrease of ATP level evoked by E260 in malignant cells, was examined after 12h of treatment since this time point precedes the onset of cell-death in the treated cells. This enabled us to examine the cellular role of Autophagy at this critical time point, when the cells are still viable but are already affected by E260 (**Figure 4K**). The effect of 3MA on the cytotoxic activity of E260 was determined after 16h treatment since this time point marks the onset of cell-death in the treated cells.

Following the reviewer's suggestion we examined the effect of the Autophagy inhibitor- 3-MA on the E260 cytotoxic activity after extended and simultaneous exposure of malignant cells to the two compounds, for 48h, 72h and 96h,. This analysis revealed that 3MA indeed delays the onset of 100% cell-death evoked by E260 in malignant cells, from 24h to 96h (Presented in the newly added **Figure S4**). We refer to this effect in the revised Discussion section.

3. Fig. 3 O-T, Fig. 4 A-B, Fig. 5 D-E and Fig. 6 C-D all utilize the difference between

compare SW620 (metastatic colon cancer) to Hfb cells. It might be possible that observed differences might result from different tissue origins rather than healthy and neoplastic. This possible bias should be addressed by providing another cell line - not for all experiments, but for some of the important ones.

Authors' response: Following the reviewer's suggestion we have extended the comparative analysis of the metabolic effects of E260 on malignant and non-malignant cells. In addition to the normal fibroblasts we have now also examined the effect of E260 on the electron transport chain (ETC) complex I activity and the mitochondria membrane potential (MMP) of the normal human epithelial CCD841CoN cells. Similarly to normal human fibroblasts, no effect of E260 was seen on the complex I activity and the MMP in epithelial CCD841CoN cells. These results are presented in the newly added **Figures 3U-X**.

4. I wonder how E260 targets only cancer cells. This should be discussed in more detail, beyond metabolism. I'd be very interested in an experiment such as an ischemic model: Does E260 induce more cell death in these "diseased" cells?

Authors' response: Following the reviewer's suggestion we have examined the effect of E260 on normal epithelial CCD841CoN cells subjected to hypoxia- a stress condition which partially imitates ischemic stress. As can be seen in the newly added **Figure S2A** no effect of E260 was observed on normoxic or hypoxic CCD841CoN cells. Thus, as we refer to in the Discussion section we assume that the selective effect of E260 on malignant cells stems from the fact that Fer/FerT associate with the mitochondrial ETC of malignant but of normal cells, thereby turning the mitochondria of malignant cells vulnerable to the effect of the Fer/FerT inhibitor-E260. The deleterious effect of E260 on the mitochondria of malignant cells combined with its other metabolic outcomes, like autophagy induction and PARP-1 activation, leads to the onset of energy crisis and necrotic death in malignant but not in normal cells.

References

1. Raucci A, Palumbo R, Bianchi ME. HMGB1: a signal of necrosis. *Autoimmunity* **40**, 285-289 (2007).

Reviewers' comments:

Reviewer #1 (Remarks to the Author):

The authors have addressed quite well the Minor Points that I had raised. However, their response to the Major Point is far from satisfactory. This is the first report of a new chemical entity in the journal literature and full characterisation data should be given, along with details of the synthetic methods. The authors should consult the standards required by a journal such as the Journal of Medicinal Chemistry. All they have provided is a synthetic scheme and a scan of a ^1H NMR spectrum. I would expect full experimental details of the synthesis, along with proper characterisation of the compound (HRMS, ^1H NMR with assignments, ^{13}C NMR, HPLC (to demonstrate purity), IR, possibly with CHN microanalysis). The manuscript should not be accepted until these are provided (at least in the Supplementary information) and reviewed.

Reviewer #2 (Remarks to the Author):

The authors have adequately addressed all concerns by this referee! They are congratulated to an important contribution to this field!

Reviewer's I comment:

- 1. Major point:** This is the first report of a new chemical entity in the journal literature and full characterization data should be given, along with details of the synthetic methods. The authors should consult the standards required by a journal such as the Journal of Medicinal Chemistry. All they have provided is a synthetic scheme and a scan of a ¹H NMR spectrum. I would expect full experimental details of the synthesis, along with proper characterization of the compound (HRMS, ¹H NMR with assignments, ¹³C NMR, HPLC (to demonstrate purity), IR, possibly with CHN microanalysis)

Authors' response: Following the reviewer's comment we include in the revised manuscript a more detailed schematic description of the synthetic steps leading to the E260 compound (Supplementary **Fig. 1 C** in the revised manuscript), and a step by step detailed synthesis process of E260 is now described in the revised **supplementary Methods** section. We have fully chemically characterized the final compound E260 by performing FTIR, ¹H-NMR, ¹³C-NMR, LCMS, and UPLC analysis coupled to high resolution mass spectrometry (HRMS) and included all the data in the revised manuscript. We also provide in the revised manuscript spectral characterization and LCMS analysis of the novel intermediates (which are chemical new entities and are not commercially available). We present in the revised **supplementary Methods** section the peak list and mass values of the intermediates and the final E260 compound. The data is presented according to the characterization format and g-policies of structurally-novel chemical compounds, as dictated by the Nature Communications journal. Detailed description of the applied analytical methods is added to the revised **supplementary Methods** section. Finally, the obtained raw data of all the E260 analyses including the UPLC profile analysis (to demonstrate purity) is included in the revised Supplementary **Fig. 1 C-I**, and the raw data of the novel chemical intermediates analyses is presented in the revised Supplementary **Fig. 2 B-D** . .

All the above described additions, and amendments are being marked in the revised manuscript.

We would like to thank the editors and reviewers for their constructive and helpful comments and suggestions. We feel that we have adequately addressed the editors' and reviewers' comments and concerns.

We believe that our findings are novel, significant, and bear general implications on the drug discovery and cancer biology and therapy fields. We therefore hope that you'll find this revised manuscript suitable for publication in the "Nature Communications" journal.

Reviewers' comments:

Reviewer #1 (Remarks to the Author):

The authors have now gone a long way to answering my previous criticisms and the manuscript is now much better, from a medicinal chemistry point of view. There only remain a few minor points which need to be resolved (in the Supplementary Information):

Compound II is novel and needs to be characterised with ¹H NMR data and evidence of elemental composition (CHN microanalysis or high-resolution MS (high-resolution to four decimal places of Da)).

Compound III is already known in the literature as the free base. It needs to be characterised with ¹H NMR data and a melting point.

Compound IV is novel needs to be characterised with ¹H NMR data and evidence of elemental composition (CHN microanalysis or high-resolution MS).

Compound Va is already known but it needs to be characterised with ¹H NMR data and measurement of the melting point. The melting point should be compared in the SI with a reported mp for this compound, e.g. mp 191-192degrees C [Raison, C. G. Preparation and reactions of thiocarbamoyl- and thioureido-amidines. *J. Chem. Soc.* 1957, 2858-2861].

Compound VI is already known but it needs to be characterised with ¹H NMR data. The mp needs to be compared with a reported value for this compound, e.g. mp 180-181 degrees C [Heindl, J.; Schröder, E.; Kelm, H. W. Chemotherapeutic nitroheterocycles. XX. Some substituted 2-nitro-1,3,4-thiadiazoles. *Eur. J. Med. Chem.* 1975, 10, 121-124]. It will also be necessary to explain the discrepancy between the observed value (193-195°C) and the literature values.

Compound VIII is novel. The melting point should be reported, along with evidence of elemental composition (CHN or HRMS).

Compound IX is novel. The melting point should be reported, along with evidence of elemental composition (CHN or HRMS).

Compound X is novel. The melting point should be reported.

Reviewer's I comments:

1. **Compound II** is novel and needs to be characterized with ¹H NMR data and evidence of elemental composition (CHN microanalysis or high-resolution MS (high-resolution to four decimal places of Da)).

Author's response: Following the reviewer's comment we have characterized **Compound II** with ¹H NMR data and high-resolution Mass Spectrometry (HRMS) (to five decimal places of Da) analysis. The actually measured and calculated molecular weights are being compared. We have also determined the melting point (MP) for this compound

2. **Compound III** is already known in the literature as the free base. It needs to be characterized with ¹H NMR data and a melting point.

Author's response: We have characterized **Compound III** with ¹H NMR data and MP analysis.

3. **Compound IV** is novel needs to be characterized with ¹H NMR data and evidence of elemental composition (CHN microanalysis or high-resolution MS).

Author's response: We have characterized **Compound IV** with ¹H NMR data and HRMS.

4. **Compound Va** is already known but it needs to be characterized with ¹H NMR data and measurement of the melting point. The melting point should be compared in the SI with a reported mp for this compound, e.g. mp 191-192degrees C [Raison, C. G. Preparation and reactions of thiocarbamoyl- and thioureido-amidines. J. Chem. Soc. 1957, 2858-2861].

Author's response: We have characterized **Compound Va** with ¹H NMR analysis and measurement of its MP. The measured MP is being compared to the MP reported for this compound.

5. **Compound VI** is already known but it needs to be characterized with ¹H NMR data. The mp needs to be compared with a reported value for this compound, e.g. mp 180-181 degrees C [Heindl, J.; Schröder, E.; Kelm, H. W. Chemotherapeutic nitroheterocycles. XX. Some substituted 2-nitro-1,3,4-thiadiazoles. Eur. J. Med. Chem. 1975, 10, 121-124]. It will also be necessary to explain the discrepancy between the observed value (193-195°C) and the literature values.

Author's response: We have characterized **Compound VI** with ¹H NMR analysis and re-measurement of its MP. The measured MP is being compared to the MP reported for

this compound in the literature. The difference between these values could stem from differences in purity and measurement methods.

- 6. Compound VIII** is novel. The melting point should be reported, along with evidence of elemental composition (CHN or HRMS).

Author's response: We have determined the MP of **Compound VIII** along with HRMS analysis .

- 7. Compound IX** is novel. The melting point should be reported, along with evidence of elemental composition (CHN or HRMS).

Author's response: We have determined the MP of **Compound IX** along with HRMS analysis.

- 8. Compound X** is novel. The melting point should be reported.

Author's response: The MP of **Compound X** has been measured and is now being reported.

All the above described additions, and amendments are included in the revised **Supplementary Information** section, and are being marked in the revised manuscript.

We would like to thank the editors and reviewers for their constructive and helpful comments and suggestions. We feel that we have adequately addressed the editors' and reviewers' comments and concerns.

We believe that our findings are novel, significant, and bear general implications on the drug discovery and cancer biology and therapy fields. We therefore hope that you'll find this revised manuscript suitable for publication in the "Nature Communications" journal.

REVIEWERS' COMMENTS:

(Reviewer #1):

The authors have addressed all the outstanding points satisfactorily and the manuscript is now suitable for publication.